# OFF-TRAJECTORY REASONING: CAN LLMS COLLABORATE ON REASONING TRAJECTORIES?

**Aochong Oliver Li, Tanya Goyal**
Department of Computer Science, Cornell University
`aochongli@cs.cornell.edu, tanyagoyal@cornell.edu`

## ABSTRACT

Reasoning LLMs are trained to verbalize their thinking process, yielding strong gains on reasoning tasks. This transparency also opens a promising direction: multiple reasoners should directly collaborate on each other's thinking on a shared trajectory, yielding better inference efficiency and exploration. A key prerequisite, however, is their abilities to assess usefulness of and build on other models' partial thinking traces – we call this *off-trajectory reasoning*. Our paper investigates a critical question: can standard *solo-reasoning* training pipelines yield desired *off-trajectory* behaviors? To this end, we propose twin tests that capture the two extremes of the spectrum: **Recoverability**, which tests whether LLMs can backtrack from "distractions" induced by misleading reasoning traces, and **Guidability**, which tests their ability to build upon correct reasoning from stronger collaborators. Our study evaluates 15 open-weight LLMs (1.5B–32B) and reveals a counterintuitive finding – "stronger" LLMs on benchmarks are often more fragile under distraction. Moreover, all models tested fail to effectively leverage guiding steps from collaborators on problems beyond their inherent capabilities, with solve rates remaining under 9.2% for math. Finally, we conduct control studies to isolate the effects of three factors in post-training on these behaviors: the choice of distillation teacher, the use of RL, and data selection strategy. Our results provide actionable insights for training natively strong reasoning collaborators; e.g., we find that sub-optimal recoverability behaviors of teacher models are transferred to distilled students even if the distilled data trajectories are correct. Taken together, this work introduces the framework for evaluating multi-model collaborations under shared reasoning, while revealing limitations of off-the-shelf reasoning LLMs.

## 1 INTRODUCTION

LLMs with thinking abilities, such as OpenAI's o-series (Jaech et al., 2024), DeepSeek-R1 (Guo et al., 2025), and Qwen3 Thinking (Yang et al., 2025a), have recently emerged as frontier models for complex reasoning tasks such as mathematics and coding. These models, trained with reinforcement learning with verifiable reward (RLVR) (Shao et al., 2024) or distillation (Hinton et al., 2015), learn to verbalize their thinking process and exhibit self-reflective behaviors (Gandhi et al., 2025). However, as these models are deployed in settings like agentic systems, their thinking traces are increasingly interleaved with tokens they did not produce – tool outputs, code execution results, or retrieved documents.

If models can already interleave their thinking with external content, can multiple reasoners directly collaborate on a shared reasoning trajectory? In this paradigm (Figure 1), a stronger model could correct a weaker one's misstep mid-derivation, or a human overseer could steer reasoning away from an unsafe path. More broadly, this has implications

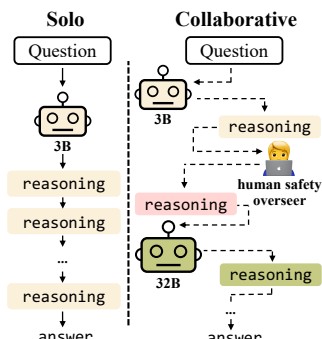

Figure 1: Comparison of solo vs. collaborative reasoning setting. On the right half, LLMs of different sizes and functionalities collaborate on a shared trajectory.

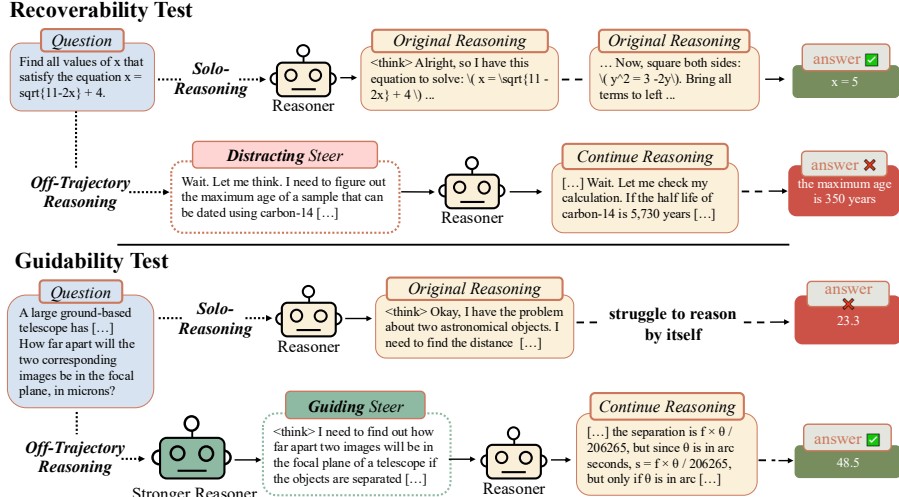

Figure 2: Illustration of the twin tests: we perturb a model's reasoning trajectories with off-trajectory steers to evaluate its *recoverability* (under a distracting steer) or *guidability* (under a guiding steer). The distracting steer is sampled from the same reasoner but for a different question.

for: (1) **Efficiency**: to balance performance and inference speed, large LLMs can focus on challenging derivations while offloading routine sub-steps (e.g., arithmetic checking) to smaller models (Akhauri et al., 2025; Yang et al., 2025c; Chen et al., 2023). (2) **Exploration**: models/humans with complementary expertise can broaden the reasoning search by spawning diverse branches (Chen et al., 2025; Qi et al., 2025; Pan et al., 2025) and composing their skills to solve cross-domain tasks. (3) **Safety**: an overseer model or human can intervene to steer ongoing reasoning in a safer direction, rather than abruptly terminating it (Wu et al., 2025; Zhang et al., 2025; Korbak et al., 2025).

Most LLMs today are trained to reason entirely on their own – what we term *solo-reasoning*. But multi-model collaboration require models to reason over trajectories interleaved with off-distribution tokens from other reasoners. We call this *off-trajectory reasoning* and ask: **can off-the-shelf solo-reasoning LLMs handle off-distribution thinking tokens in their reasoning trajectories?**

We approach this question by decomposing off-trajectory reasoning into two complementary tests, **Recoverability** and **Guidability**, and evaluating both in simulated collaboration scenarios (see Figure 2). The recoverability test evaluates whether LLMs can backtrack from misleading reasoning injected into their trajectory to continue their original correct reasoning. At the other end of the spectrum, the guidability test evaluates whether LLMs can build upon correct yet incomplete reasoning from guiding models to tackle problems they are unable to solve via solo-reasoning.[1]

We systematically evaluate 15 open-weight LLMs across math and coding benchmarks (MAA, 2024; 2025; Hendrycks et al., 2021; Lewkowycz et al., 2022; He et al., 2024; Gu et al., 2024; Chen, 2021; Austin et al., 2021; Liu et al., 2023). Counterintuitively, benchmark performance does not predict off-trajectory robustness: `AM-Thinking-32B`, the top math model (82.6% average), recovers only 33.4% of the time, whereas `Qwen3-1.7B` (59.9%) recovers 98.4%. Overall, recoverability falls to 74.9% for math and 59.1% for coding on problems originally solved correctly. The guidability test reveals a similar ceiling – no model exceeds 9.2% on math, and while the same models show higher apparent guidability (up to 47.3%), most of the guiding reasoning already contains the answers (§E.3). These results show that benchmark optimization does not account for off-trajectory reasoning capabilities.

We further investigate how post-training decisions shape off-trajectory behaviors through controlled experiments on three factors: distillation teacher choice, RL training, and data selection strategy. Most strikingly, we find that the recoverability weaknesses of teacher models transfer to distilled students even when training uses only the teacher's correct trajectories. We also show that RL

---

[1]We systematically test for correctness of reasoning in this paper. However, our framework can be extended for other aspects of alignment, e.g., can solo LLMs robustly reject unsafe collaborator trajectories?

can substantially improve off-trajectory robustness where supervised fine-tuning saturates, and that aggressive data filtering can introduce high variance in recoverability across training checkpoints.

In summary, our work makes the following contributions:

1. We introduce the **Recoverability** and **Guidability** tests as a systematic framework for evaluating off-trajectory reasoning. Our setup complements existing standard solo-reasoning benchmarks by offering an orthogonal view into model reasoning capabilities. (§2)

2. Equipped with this framework, we evaluate 15 open-weight LLMs for off-trajectory reasoning. Our analysis reveals key limitations of "strong" solo reasoners and shows that almost all models fail at exploiting correct guidance to improve beyond their inherent capability limits. (§3)

3. We conduct the first controlled study isolating the **direct effects of post-training decisions** on off-trajectory behaviors. Our experiments reveal that a teacher's recoverability weaknesses transfer to distilled students, that RL yields substantial recoverability gains where SFT plateaus, and that "less-is-more" data filtering introduces high variance in off-trajectory robustness. (§4)

## 2 TWIN TESTS FOR OFF-TRAJECTORY REASONING

**Preliminaries and Notations.** Let $M$ be a reasoning model and $(q, a^*)$ be a training or test data point. In standard solo-reasoning, $M$ generates a reasoning trajectory $r = [r_1, r_2, r_3, \ldots]$, a sequence of *reasoning units*, and a final answer $a$ for an input question $q$, i.e. $(r, a) \sim M(\cdot|q)$. The granularity of a reasoning unit can be flexibly determined.

In contrast, in the collaborative setting, multiple models or different instantiations of the same model contribute different parts to the reasoning trajectory $\mathbf{r}$. Recent work has explored some collaboration strategies, such as dynamically off-loading reasoning sub-parts to weaker/stronger models (Yan et al., 2025; Zhou et al., 2025; Akhauri et al., 2025; Yang et al., 2025c), orchestrating meta-thinking and reasoning agents (Wan et al., 2025), tooling (Jin et al., 2025), injecting control tokens to extend reasoning (Muennighoff et al., 2025), or aggregating parallel samples (Zhao et al., 2025; Qi et al., 2025) during both training and inference time.

The success of such collaboration hinges on the main model $M$'s ability to process and build upon a trajectory mixing both in- and off-distribution reasoning units $\mathbf{r} = [r^M, r^{M'}, r^{M''}, \ldots]$. In this paper, we instantiate a simplified setup of two-model collaboration to probe off-trajectory reasoning capabilities in frontier open-weight LLMs.

**Two-model Setup.** We simulate a collaboration between two reasoning systems, where the main model $M$ and the collaborator $M_{\text{steer}}$ jointly contribute to an off-trajectory reasoning $[r^{\text{og}}, r^{\text{steer}}]$. In practice, we construct $r^{\text{og}}$ by sampling from the main model $M$ and stopping generation at $m$ tokens, i.e. $|r^{\text{og}}| = m$. Similarly, $r^{\text{steer}}$ is sampled from the collaborator with $|r^{\text{steer}}|$ limited to $n$ tokens. To measure off-trajectory reasoning performance, we concatenate these two incomplete trajectories to construct a shared off-distribution trajectory. Finally, we sample a reasoning completion and final answer from $M$ conditioned on the original question and this trajectory.

$$(\mathbf{r}^{\text{off}}, a^{\text{off}}) \sim M(\cdot \mid q, [r^{\text{og}}, r^{\text{steer}}])$$

For domains with verifiable rewards, we measure success as the accuracy of the final answer $a^{\text{off}}$ against the ground truth $a^*$.

**Considerations for designing the steer.** This simplified setup enables us to flexibly and scalably test how $M$ behaves when $r^{\text{steer}}$ falls at two extremes. We design twin tests: (i) **Recoverability**, which tests whether LLMs can resist a incorrect and distracting steer, and backtrack to previous reasoning, and (ii) **Guidability**, which tests models' abilities to successfully leverage a guiding steer to surpass its solo-reasoning ability.

These twin tests differ mainly in two aspects: (1) **the selection of test questions**, $q$ and (2) **the construction of steered trajectories $[r^{\text{og}}, r^{\text{steer}}]$**. Given an original test set and test model $M$, our protocol automatically constructs $M$-specific off-trajectory datasets for both tests, each consisting of $(q, [r^{\text{og}}, r^{\text{steer}}], a^*)$. The overall process for this is shown in Figure 2 and described below.

## 2.1 RECOVERABILITY TEST

**Selecting test datapoints $\{(q, a^*)\}$.** For a given test model $M$, we select the subset of test questions that $M$ can correctly answer in solo-reasoning, i.e. $a = a^*$, where $(r, a) \sim M(.|q)$. This disentangles the effects of distracting steers from $M$'s inherent capabilities.

**Constructing steered trajectories.** The trajectory consists of two parts: $r^{\text{og}}$ and $r^{\text{steer}}$. We truncate $r$, the reasoning trajectory from solo-reasoning, to the first $m$ tokens to obtain $r^{\text{og}}$, always at the end of the nearest sentence to preserve coherence.

We require $r^{\text{steer}}$ to be a strong distractor for the test model $M$. However, it is difficult to *a priori* determine which model $M_{\text{steer}}$ and steer $r^{\text{steer}}$ will achieve this reliably. Therefore, we simulate the distraction $r^{\text{steer}}$ by sampling from $M$ itself, but conditioned on a different question $q'$. So, if $M$ is distracted to blindly complete $r^{\text{steer}}$, its reasoning is then guaranteed to be incorrect. In practice, we control the length of $r^{\text{steer}}$ by truncating it to the first $n$ tokens of $r'$, where $(r', a') \sim M(.|q')$. In our experiments, we control the strength of the distractor by varying $n$, i.e. $|r^{\text{steer}}|$ and the insertion point by varying $m$, i.e. $|r^{\text{og}}|$. We describe procedure details and design rationale in Appendix E.1.

## 2.2 GUIDABILITY TEST

**Selecting test datapoints $\{(q, a^*)\}$.** We select the subset of test questions for which the solo-reasoning solve-rate is either $0$ or $1$ out of $8$ samples. These are problems $M$ cannot reliably solve on its own, so any improvement must come from the help of guiding steers.

**Constructing steered trajectories.** First, unlike the recoverability test, we do not include $M$'s own partial reasoning $r^{\text{og}}$ (i.e., set $m = 0$): the guide's steer is placed at the very beginning of the thinking, before $M$ has produced any tokens. This is because $r^{\text{og}}$ might already contain errors that anchor $M$ to a wrong direction, thereby confounding the measurement of guidability.

We construct $r^{\text{steer}}$ using a stronger reasoner $M_{\text{steer}}$ as the guide, i.e. with a higher benchmark performance than $M$. Figure 2 illustrates this. To test whether $M$ can build on $M_{\text{steer}}$'s correct reasoning, we only provide the first $n$ tokens of the complete trajectory. In practice, we vary the "amount" of guidance by varying $n$ to different fractions of the complete trajectory from the guide. Moreover, we use multiple guiding models $M_{\text{steer}}$ to construct independent steers for each $q$. This allows us to measure guidability under different steer distributions and amount of guidance.

## 3 OFF-THE-SHELF EVALUATION & RESULTS

### 3.1 EXPERIMENT SETUP

**Models, Datasets, and Benchmarks.** We run our experiments on 15 open-weight models. To illustrate the relationships between these LLMs, we group them into four families (see Figure 3):

- **DeepSeek-R1** (Guo et al., 2025): `R1-Qwen-1.5B/7B/32B` and `R1-Llama-8B` are distilled from `DeepSeek-R1` using supervised fine-tuning (SFT).
- **Qwen3** (Yang et al., 2025a): `Qwen3-32B` is trained using RL for reasoning without distillation, while `Qwen3-1.7B/8B/30B-A3B` are distilled from `Qwen3-235B` and `Qwen3-32B`.
- **QwQ**: `QwQ-32B` (Qwen Team, 2025) is trained from `Qwen2.5-32B-Base` model with RL. `OpenThinker3-1.5B/7B` (Guha et al., 2025) are based on `Qwen2.5-Instruct` and distilled from `QwQ-32B` on 1.2M curated math and coding examples.
- **Community**: `DeepScaleR-1.5B` (Luo et al., 2025) and `DeepMath-1.5B` (He et al., 2025b) are trained using RL on `R1-Qwen-1.5B` using DeepScaleR and DeepMath datasets respectively. `LIMO-32B` (Ye et al., 2025) is SFT from `Qwen2.5-32B-Instruct` on the LIMO dataset of 817 examples. Finally, `AM-Thinking-32B` (Ji et al., 2025) is a `Qwen2.5-32B-Base` model first distilled on 2.84M examples and RL on 54K math and coding questions.

We evaluate on $1,507$ math questions from five benchmarks: AIME-2024 (MAA, 2024), AIME-2025 (MAA, 2025), MATH-500 (Hendrycks et al., 2021), Minerva Math (Lewkowycz et al., 2022), and OlympiadBench (He et al., 2024). For code, we evaluate on $1,762$ questions from four benchmarks: CruxEval (Gu et al., 2024), HumanEval (Chen, 2021), MBPP (Austin et al., 2021), and EvalPlus (Liu et al., 2023).

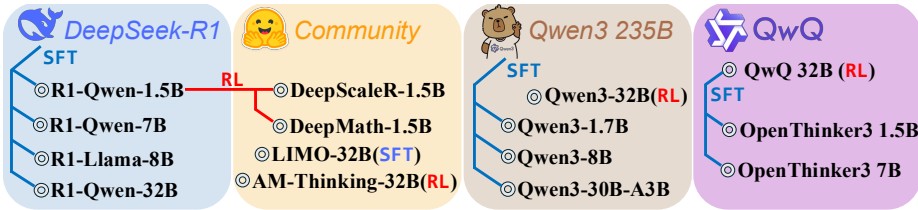

Figure 3: 15 open-weight LLMs grouped into four families. The branches indicate the source from which LLMs are derived, and the colors indicate SFT/RL training methods.

**Hyperparameter Settings.** All LLMs are evaluated under the same hyperparameter settings: a maximum of 32K tokens, temperature 0.6, top-$p$ 0.95, and no system prompt. For each question, we sample 8 completions and report the average Pass@1.

**Recoverability and guidability Setup.** Following the protocols in §2.1, we sample 200 original trajectories $r^{og}$ and 50 trajectories as distracting steers ($r^{steer}$) for each LLM. By default, we set $n$, i.e., $|r^{steer}|$, to be 0.2 times the original length of the distracting trajectory; this leaves sufficient tokens for *off-trajectory* completion. We set $m$, i.e., $|r^{og}|$, to be 0, 0.2, 0.4, 0.6, and 0.8 times the length of the original reasoning from the main model. We report recoverability on two subsets: (1) *shared* subset that includes questions that all 15 LLMs can fully solve (8 out of 8), and (2) *individual* subset that samples questions independently for each LLM following the criterion defined in §2.1.

We instantiate the guidability tests using `DeepSeek-R1`, `Qwen3-235B`, and `QwQ-32B` as $M_{steer}$ to sample *guiding* steers $r^{steer}$. Since the best 5 LLMs almost saturate the benchmarks, we only evaluate on the remaining 10 LLMs that have enough questions with solve rate $\leq 1/8$ (Table 13). We set $n$, i.e., $|r^{steer}|$, to be 0.2, 0.4, 0.6, and 0.8 times the total tokens in the guide's reasoning. Similar to the recoverability test, we report guidability scores on two subsets: *shared* (intersection across models) and *individual* (per model).

## 3.2 RESULTS

Our main results are shown in Table 1. We group models into low, medium, and high tiers based on their solo-reasoning performance (reported in the *Avg. Benchmark* column) and report recoverability and guidability results on both shared and individual subsets.

**Finding 1: Stronger solo-reasoners $\neq$ stronger collaborators.** Surprisingly, we find that recoverability is largely orthogonal to LLMs' solo-reasoning performance. Particularly, we highlight models in the *low* benchmark tier for math, such as `OpenThinker3-1.5B` and `Qwen3-1.7B`, exhibit substantially better recoverability than *medium* and *high* tier models like `QwQ-32B` and `Qwen3-32B`. Noticeably, the best performing solo-reasoning math model `AM-Thinking-32B` reports the second worst recoverability performance. Similarly, `LIMO-32B` – claimed to surpass prior SFT approaches using only 1% of training data – recovers less than 30% of the time on the math benchmarks. Across models, the average recoverability on the shared math subset is 74.9%, a 25.1 percentage-point drop from the original 100% solve rate, when their trajectories are perturbed with tangential distractions.

We observe similar trends for code in Table 1. The three best performing models in the *high* tier (`R1-Qwen-32B`, `AM-Thinking-32B` and `QwQ-32B`) report low recoverability rates (in the range $40 - 48\%$ range), while three out of four models in the *low* tier report a greater than 65% recoverability rate on the shared test subset. Combined, these findings on both math and code provide evidence that **models optimized heavily for popular benchmarks may have hidden vulnerabilities, particularly in off-trajectory reasoning**.

**Finding 2: The invisible guidability ceiling.** Our results show that all LLMs report exceptionally low guidability scores for math; none of the models report $> 10\%$ on the shared subset for math (and at most 35.0% on the individual subset). This indicates that all math models, regardless of their solo-reasoning capabilities, struggle to effectively build upon guiding trajectories. Crucially, the performance does not improve even when models are paired with their own distillation teacher, i.e., the model whose samples they were trained on (see Table 16 for full set of results). For example,

| Model | Family | Benchmark Avg. | Recoverability Sh. | Ind. | Guidability Sh. | Ind. |
|---|---|---|---|---|---|---|
| **(a) Math** | | | | | | |
| *Low Benchmark Scores* | | | | | | |
| R1-Qwen-1.5B | DS-R1 | 47.5 | $60.6_{\uparrow+2}$ | $38.6_{\uparrow+2}$ | $3.0_{\uparrow+0}$ | $28.4_{\uparrow+5}$ |
| R1-Llama-8B | DS-R1 | 54.1 | $81.4_{\uparrow+6}$ | $49.6_{\uparrow+4}$ | $8.7_{\uparrow+5}$ | $35.0_{\uparrow+8}$ |
| DeepMath-1.5B | Comm. | 54.8 | $88.0_{\uparrow+10}$ | $61.8_{\uparrow+7}$ | $3.4_{\downarrow-1}$ | $27.1_{\uparrow+2}$ |
| DeepScaleR-1.5B | Comm. | 55.3 | $82.4_{\uparrow+5}$ | $52.9_{\uparrow+3}$ | $4.1_{\downarrow-1}$ | $29.8_{\uparrow+3}$ |
| OpenThinker3-1.5B | QwQ | 59.2 | $95.2_{\uparrow+9}$ | $71.8_{\uparrow+8}$ | $5.7_{\downarrow-1}$ | $32.7_{\uparrow+4}$ |
| Qwen3-1.7B | Qwen3 | 59.9 | $98.4_{\uparrow+9}$ | $74.6_{\uparrow+9}$ | $6.1_{\uparrow+0}$ | $29.9_{\uparrow+2}$ |
| *Medium Benchmark Scores* | | | | | | |
| R1-Qwen-7B | DS-R1 | 64.6 | $73.5_{\downarrow-1}$ | $45.8_{\downarrow-2}$ | $6.0_{\downarrow-2}$ | $19.7_{\downarrow-6}$ |
| LIMO-32B | Comm. | 67.3 | $29.3_{\downarrow-7}$ | $18.5_{\downarrow-7}$ | $8.8_{\uparrow+0}$ | $21.5_{\downarrow-5}$ |
| OpenThinker3-7B | QwQ | 72.1 | $85.6_{\uparrow+1}$ | $74.5_{\uparrow+5}$ | $9.1_{\uparrow+0}$ | $20.6_{\downarrow-7}$ |
| R1-Qwen-32B | DS-R1 | 72.3 | $69.8_{\downarrow-6}$ | $45.6_{\downarrow-6}$ | $9.2_{\uparrow+0}$ | $22.5_{\downarrow-6}$ |
| *High Benchmark Scores* | | | | | | |
| Qwen3-8B | Qwen3 | 79.1 | $85.9_{\uparrow+0}$ | $68.8_{\uparrow+1}$ | N/A | N/A |
| QwQ-32B | QwQ | 80.5 | $79.7_{\downarrow-5}$ | $62.6_{\downarrow-1}$ | N/A | N/A |
| Qwen3-32B | Qwen3 | 81.0 | $71.8_{\downarrow-8}$ | $56.9_{\downarrow-5}$ | N/A | N/A |
| Qwen3-30B-A3B | Qwen3 | 81.1 | $87.8_{\downarrow-2}$ | $60.0_{\downarrow-5}$ | N/A | N/A |
| AM-Thinking-32B | Comm. | 82.6 | $33.4_{\downarrow-13}$ | $25.3_{\downarrow-13}$ | N/A | N/A |
| **(b) Coding** | | | | | | |
| *Low Benchmark Scores* | | | | | | |
| R1-Qwen-1.5B | DS-R1 | 53.3 | $40.6_{\uparrow+0}$ | $28.7_{\uparrow+0}$ | $24.1_{\uparrow+2}$ | $40.1_{\uparrow+2}$ |
| OpenThinker-1.5B | QwQ | 53.4 | $78.2_{\uparrow+12}$ | $53.2_{\uparrow+8}$ | $35.8_{\uparrow+2}$ | $49.4_{\uparrow+2}$ |
| DeepScaleR-1.5B | Comm. | 58.8 | $65.1_{\uparrow+6}$ | $46.2_{\uparrow+4}$ | $23.2_{\downarrow-1}$ | $37.3_{\downarrow-1}$ |
| DeepMath-1.5B | Comm. | 59.8 | $74.6_{\uparrow+7}$ | $56.3_{\uparrow+7}$ | $20.9_{\downarrow-3}$ | $36.2_{\downarrow-3}$ |
| *Medium Benchmark Scores* | | | | | | |
| Qwen3-1.7B | Qwen3 | 74.7 | $77.4_{\uparrow+8}$ | $61.4_{\uparrow+8}$ | $47.3_{\uparrow+2}$ | $56.4_{\uparrow+2}$ |
| R1-Llama-8B | DS-R1 | 76.0 | $56.1_{\uparrow+1}$ | $46.2_{\uparrow+2}$ | $47.2_{\uparrow+0}$ | $54.4_{\uparrow+0}$ |
| R1-Qwen-7B | DS-R1 | 77.7 | $54.5_{\downarrow-1}$ | $37.4_{\downarrow-3}$ | $45.7_{\downarrow-2}$ | $53.1_{\downarrow-2}$ |
| OpenThinker-7B | QwQ | 80.1 | $74.9_{\uparrow+4}$ | $68.2_{\uparrow+6}$ | N/A | N/A |
| *High Benchmark Scores* | | | | | | |
| Qwen3-8B | Qwen3 | 87.4 | $67.6_{\uparrow+1}$ | $60.9_{\uparrow+3}$ | N/A | N/A |
| Qwen3-32B | Qwen3 | 87.8 | $46.8_{\downarrow-6}$ | $30.8_{\downarrow-8}$ | N/A | N/A |
| Qwen3-30B | Qwen3 | 89.0 | $59.6_{\downarrow-3}$ | $47.7_{\downarrow-2}$ | N/A | N/A |
| R1-Qwen-32B | DS-R1 | 89.7 | $40.7_{\downarrow-10}$ | $34.8_{\downarrow-9}$ | N/A | N/A |
| AM-Thinking-32B | Comm. | 89.7 | $43.0_{\downarrow-9}$ | $40.7_{\downarrow-7}$ | N/A | N/A |
| QwQ-32B | QwQ | 90.5 | $47.9_{\downarrow-9}$ | $44.2_{\downarrow-8}$ | N/A | N/A |

Table 1: **Recoverability and guidability results for math (a) and coding (b).** Columns report benchmark averages and recoverability/guidability scores for *shared* (Sh.) and *individual* (Ind.) subsets. Models are grouped into low/medium/high tiers by *Benchmark Avg*. Subscripts indicate rank changes relative to the benchmark ranking ($+k$ rise, $-k$ drop); green ($\uparrow$) denotes improvement, red ($\downarrow$) decline. "DS-R1" = DeepSeek-R1 family, "Comm." = Community models. N/A = not evaluated. Across both domains, benchmark performance is largely orthogonal to recoverability—the best benchmark model in each panel exhibits among the worst recoverability.

`Qwen3-1.7B` shows no guidability gains when guided by `Qwen3-235B` compared to other models. We observe better guidability results on the coding benchmarks, with most models reporting a $20-50\%$ improvement with external guidance, roughly corresponding to their solo performance.

To understand the failure modes in the guidability setting for math, we run further investigations. We find that **even these low guidability scores on math are artificially inflated.** Since we truncate the guiding steer at different lengths, it is possible that some partial $r_{\text{steer}}$ already contain the correct

| Model | Teach. (%) | Ans.? (%) | $\Delta$ |
|---|---|---|---|
| R1-Qwen-1.5B | 28.4 | 25.6 | 2.8 |
| DeepScaleR-1.5B | 29.8 | 23.3 | 6.5 |
| R1-Llama-8B | 35.0 | 21.8 | 13.2 |
| DeepMath-1.5B | 27.1 | 22.9 | 4.2 |
| OpenThinker3-1.5B | 32.7 | 26.9 | 5.8 |
| Qwen3-1.7B | 29.9 | 18.0 | 11.9 |
| R1-Qwen-7B | 19.7 | 12.1 | 7.6 |
| LIMO-32B | 21.5 | 10.2 | 11.3 |
| OpenThinker3-7B | 20.6 | 13.8 | 6.8 |
| R1-Qwen-32B | 22.5 | 11.2 | 11.3 |
| Avg. | 26.7 | 18.6 | 8.1 |

Table 2: Analysis of guidability results. Teach. = guidability score (individual); Ans.? = fraction of steers already containing the correct answer; $\Delta$ = Teach. − Ans. (pp).

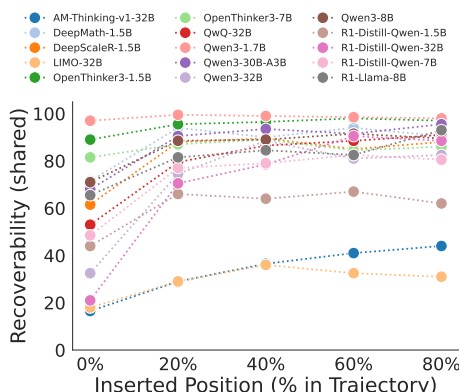

Figure 4: Recoverability (shared) across positions (%) of the original trajectory.

answer derivation. In such cases, we expect the guidability test to be trivially easy. In Table 2, we report the percentage of guiding steers that already contain the correct answer (Ans.? column). We find that this is true for 18.6% of steers on average (see Table 14 for breakdown by steer length). However, we find that LLMs can often fail to recognize such correct reasoning, reject the given answer and pivot to an incorrect path, resulting in the low guidability scores. This suggests that conditioning models on correct but out-of-distribution traces does not enable them to successfully leverage them and surpass their inherent capability limits.

**Finding 3: The beginning of model reasoning is critical for recovery.** To better understand the recoverability trends in Table 1, we visualize the recovery rates separately for different percentages (%) of the original thinking trajectory where the distracting steer is inserted. Figure 4 shows these results.[2] Interestingly, we observe a consistent pattern across models – distraction at the very start (0%) of the trajectory leads to the largest degradation. In particular, the degradation at 0% is substantially higher than at 20% position. This is surprising as models tend to restate the question at the opening stage and rarely include substantive problem solving. Given these results, we hypothesize that re-stating the question at the start is critical for models to anchor later reasoning.

To test our hypothesis, we conduct an ablation that re-instantiates the recoverability tests but preserves the first paragraph of the original trajectory. We find that most LLMs exhibit noticeable improvements across positions after this change, especially at the 0% position (complete set of results are included in Table 7 in the Appendix). In fact, the average recoverability score exceeds 83.5% for all models with this small tweak, with the exception of `LIMO-32B` and `AM-Thinking-32B`, which improve but remain below 60% – consistent with their broader recoverability weaknesses identified in Finding 1. This clearly shows that **while re-statement of the problem does not add new information, it is critical for LLM off-trajectory reasoning**.

## 4 CONTROL STUDY ON POST-TRAINING DECISIONS

Section 3 reveals that models with similar benchmark scores can have dramatically different off-trajectory robustness. However, the LLMs used in our experiments in Section 3 differ in base models, training data, and post-training recipes. Therefore, the previous results alone cannot pinpoint what drives these differences.

To disentangle these factors, we conduct controlled experiments that isolate the effects of (1) teacher models used for distillation in §4.1, (2) conducting RL based post-training after SFT in §4.2, and (3) quality heuristics for data filtering in §4.3. We conduct all our experiments in this section on the math benchmarks, as most of the small open-source models are specialized for math.

### 4.1 HOW DO TEACHERS' BEHAVIORS AFFECT DISTILLED MODELS?

**Hypothesis.** We observe in Table 1 that LLMs distilled from `DeepSeek-R1` generally have lower recoverability scores compared to those from `QwQ` and `Qwen3`. This is despite the fact that most of

---

[2]The full set of results for both shared and individual metrics are reported in Tables 6 and 8 in the Appendix.

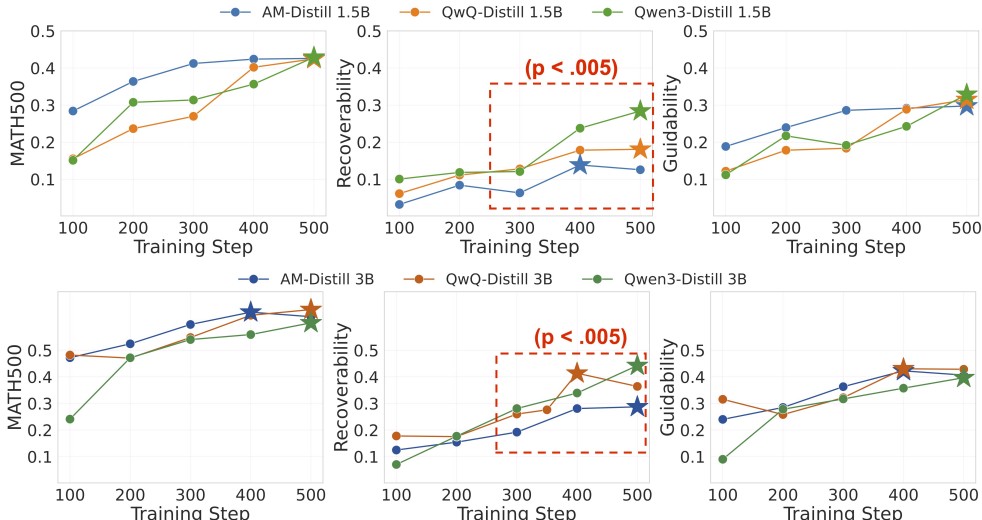

Figure 5: Qwen2.5 models (1.5B and 3B) distilled from `AM(-Thinking)-32B` show consistently lower recoverability than those distilled from `QwQ-32B` or `Qwen3-32B`, while having similar performance on benchmark and guidability; the gap is significant after step 300 (p ≤ .005). Stars mark each model's peak over training steps.

them are trained from similar base models using distillation. Therefore, we ask – *do distilled models inherit the vulnerabilities of their teachers' off-trajectory behaviors through distillation?*

**Setup**  We conduct our experiment with three LLMs as the distillation teacher models: `AM(-Thinking-32B)`, `QwQ(-32B)`, `Qwen3(-32B)`. We choose the `AM` model since it has a similarly high benchmark performance as `QwQ` and `Qwen3` models in Table 1, but significantly lower recoverability. We supervised fine-tune two `Qwen2.5` models (1.5B and 3B) on correct trajectories from each teacher separately (more details in Appendix G).

We evaluate the distilled models (`AM-/QwQ-/Qwen3-Distill 1.5B/3B`) on MATH500 for benchmark performance and twin tests. We report results for different checkpoints during training in Figure 5; we report the overall benchmark (leftmost column), recoverability (middle column) and guidability (rightmost column) performances. Additionally, we highlight checkpoints with significant differences ($p \leq 0.005$) based on two-sample t-tests.

**Results: Students mirror their teacher's recoverability performance.**  Our results in Figure 5 show that `AM-Distill` models, i.e. models trained on trajectories from `AM-Thinking-32B`, show significantly lower recoverability than `QwQ-` and `Qwen3-Distill` counterparts after step 300, despite similar benchmark and guidability scores. This recoverability gap persists across all model sizes that we tested and also remains consistent at different positions of the reasoning trajectories (Appendix G). We highlight that these surprising recoverability trends mirror the trends observed for the corresponding teacher models in Table 1.

Crucially, all teachers are distilled using only their *correct* trajectories – yet the recoverability gap persists. This means that **hidden vulnerabilities of teachers propagate through distillation even when the training data contains no errors**, implying that vulnerability is encoded in the reasoning style, not in the correctness of individual solutions. Therefore, our twin tests can serve as an additional criterion for selecting distillation teachers beyond correctness alone.

## 4.2 CAN RL FURTHER IMPROVE OFF-TRAJECTORY REASONING AFTER SFT SATURATES?

**Hypothesis.**  In Table 1, we do not observe a consistent advantage of RL over SFT distillation on twin tests. However, training recipes of these models are different, making it impossible to draw concrete conclusions about RL's impact. Here, we ask – *can RL further improve both recoverability and guidability even after SFT has saturated?*

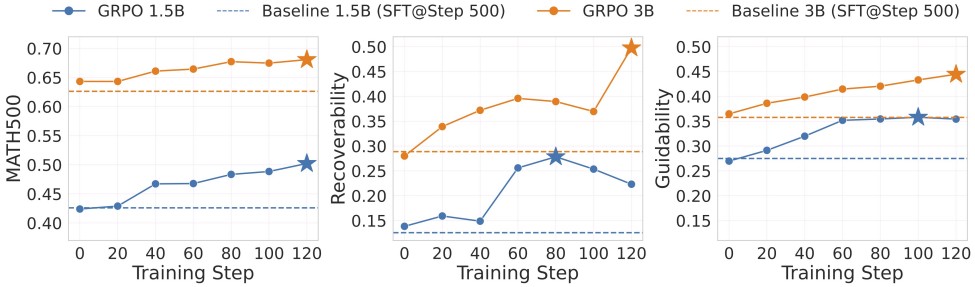

Figure 6: GRPO 1.5B and 3B (from SFT@Step 400) show noticeable gains on benchmark, recoverability, and guidability compared to the initial checkpoint and baselines (SFT@Step 500). This improvement is consistent over RL training. Stars mark the peak values over training steps.

**Setup.** We use distillation checkpoints from Section 4.1 – AM-Distill 1.5B and 3B models at step 400 – as the initial policy for RL training. This choice is motivated by: (1) we observe SFT saturates on benchmarks and twin tests after step 400; and (2) AM-Distill is shown to perform poorly in recoverability, making it more suitable to test the effects of RL. We train both models on MATH8K dataset with Grouped Relative Policy Optimization (GRPO) (Shao et al., 2024).

**Results: RL training yields substantial improvements in recoverability.** Figure 6 shows the impact of RL training on benchmark scores, recoverability and guidability. While all scores improve with RL, we see a noticeably high recoverability improvement (e.g. 15.3%-28.9%) accompanying a slight increase on benchmark scores (5.4%-7.6%) and guidability (8.3%-8.7%). Notably, RL training completely bridges the gap in recoverability that we observed in Figure 5 between AM-Distill and QwQ-/Qwen3-Distill models. We hypothesize that the key difference is what each method exposes the model to: **SFT trains exclusively on successful demonstrations, teaching what correct reasoning looks like; RL, by contrast, exposes models to failed trajectories and explicitly rewards recovery, teaching what to do when reasoning goes wrong.** We leave a more thorough investigation of the mechanisms behind observed improvement to future work.

## 4.3 DOES LESS DATA ALWAYS LEAD TO POORER RECOVERABILITY?

**Hypothesis.** Recent works have shown that data quality is critical for strong reasoning capabilities (Dang & Ngo, 2025; Albalak et al., 2025; Guha et al., 2025). The "Less-Is-More" (LIMO) hypothesis (Ye et al., 2025) pushes for an extreme version of this claim – minimal amount of "high-quality" data is sufficient to elicit reasoning. (Ye et al., 2025) curates the LIMO dataset of 817 examples filtered based on heuristics and support their claim with the performance of `LIMO-32B` model on popular reasoning benchmarks. However, we observed a contrary result in Table 1 where the `LIMO-32B` model reported the worst recoverability despite decent solo-reasoning performance. To understand this, we ask – *is less-is-more paradigm inherently limited for off-trajectory reasoning?*

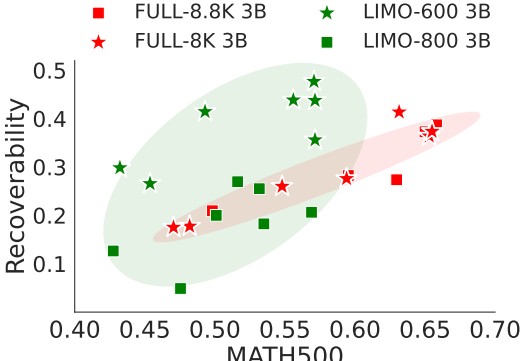

Figure 7: LIMO-600/-800 3B models exhibit greater variance in recoverability than FULL-8K/8.8K 3B. Colors: FULL, LIMO. Markers: square = contains data from LIMO-800, star = otherwise. We observe that model checkpoints trained on high-quality but limited quantity of data show high variance in recoverability scores across similar benchmark score values.

**Setup.** We train `Qwen2.5-3B-Base` models on two larger datasets of mixed "quality" and two smaller ones of only "high-quality" data: (1) **FULL-8K**: MATH8K dataset distilled from QwQ 32B in §4.1, (2) **FULL-8.8K**: a mix of FULL-8K and the LIMO dataset. (3) **LIMO-800**: the LIMO dataset, and (4) **LIMO-600**: 600

"challenging" examples we extracted from following the "LIMO" principle, i.e. classified as Level-5 difficulty and with long reasoning trajectories. Figure 7 plots recoverability scores against benchmark scores at different checkpoints during training.

**Results: Models trained on less data are not necessarily worse on recoverability but exhibit extremely high variance between checkpoints.** For example, we observe that `LIMO-600` and `LIMO-800` 3B models show markedly different levels of recoverability against similar benchmark scores. On the other hand, FULL-8K and FULL-8.8K models trained on larger datasets have minimal variance across checkpoints with the same benchmark scores.

Our results emphasize that "over-optimizing" benchmarks through data filtering could introduce unwanted biases in off-trajectory, or any off-distribution, reasoning while not being captured by standard solo-reasoning evaluations. Our tests can complement existing criteria for selecting checkpoints with higher robustness to out-of-distribution scenarios.

## 5 RELATED WORK

**Reasoning LLMs**. Recent post-training advances have led to substantial improvements on math and coding benchmarks (Huang & Yang, 2025; Guo et al., 2025), with both closed- and open-source reasoning LLMs emerging since OpenAI's o-1 (Jaech et al., 2024), e.g. Guo et al. (2025); Yang et al. (2025a); Guha et al. (2025); Ye et al. (2025); Ji et al. (2025). These models are trained to produce extended reasoning traces using RL algorithms such as PPO (Schulman et al., 2017), GRPO (Shao et al., 2024), or their variants with verifiable rewards. At smaller scales, models are primarily trained with distillation (Hinton et al., 2015). We analyze fifteen representative open-weight reasoning LLMs spanning diverse families and training paradigms.

**LLM Reasoning Intervention and Collaboration**. Recent studies intervene on LLM reasoning to understand and control their behaviors, including perturbing intermediate steps to examine their faithfulness (Arcuschin et al., 2025; Baker et al., 2025), improving instruction following and alignment behaviors (Wu et al., 2025; Muennighoff et al., 2025), and interpreting (Lee et al., 2025b; Marjanović et al., 2025) and stress-testing cognitive behaviors (Gandhi et al., 2025). Wen et al. (2025) examine the impact of thinking patterns on outcome correctness, while He et al. (2025a); Lee et al. (2025b) systematically categorize different types of reasoning strategies and errors. In closely related work, He et al. (2025a) investigate LLMs' ability to recover from unhelpful thoughts. Our work differs in three ways: (i) our distractors are sampled from models rather than hand-designed, making them naturally model-specific; (ii) we introduce guidability as a complementary test of leveraging correct off-trajectory guidance; and (iii) we trace fragility to specific training choices through controlled experiments.

Our work is also closely related to hybrid parallel and serialized scaling approaches (Pan et al., 2025), including offloading challenging reasoning parts to larger models Akhauri et al. (2025); Yang et al. (2025c) and orchestrating different models for high-level planning and downstream reasoning Lee et al. (2025a); Wan et al. (2025). In contrast to these systems, we evaluate how solo-reasoning LLMs fail when asked to continue from a shared reasoning trajectory.

## 6 CONCLUSION

In this work, we investigate off-trajectory reasoning in LLMs – their ability to think on trajectories steered by other reasoners. Through our twin tests – Recoverability and Guidability – we evaluate 15 open-weight reasoning LLMs and find that standard benchmark performance does not predict off-trajectory robustness: models that excel at solo reasoning can struggle to recover from erroneous steers or leverage correct guidance from collaborators.

Our controlled experiments reveal that these weaknesses are not incidental but directly shaped by training decisions – teacher choice propagates hidden vulnerabilities through distillation, RL improves robustness where SFT saturates, and data filtering trades stability for variance. Together, these findings show that off-trajectory robustness must be explicitly accounted for during training; it does not emerge as a byproduct of benchmark optimization.

## 7 ACKNOWLEDGMENTS

This project was partially supported by NSF grant IIS-2433072, and a gift from Google. We gratefully acknowledge use of the research computing resources of the Empire AI Consortium, Inc, with support from Empire State Development of the State of New York, the Simons Foundation, and the Secunda Family Foundation.

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

## A   LARGE LANGUAGE MODEL USAGE

In this paper, we use AI with great caution for polishing the language of some texts that are originally written by the authors.

## B   LLM-AS-A-JUDGE PROMPT

```
### System Prompt
You are an unbiased examiner who evaluates whether a student's
answer to a given question is correct.
Your task is to determine if the student's final answer matches
the standard answer provided, based solely on correctness and the
question's specific requirements.
Do not perform any additional calculations or reinterpret the
question.  Simply compare the student's answer to the standard
answer to determine if it satisfies the question's requirements.

Focus strictly on:
1.  Understanding the exact requirement of the question.
2.  Comparing the student's final answer directly and rigorously
to the provided standard answer.
3.  Your task is not to solve the problem but to determine
whether the student's answer is correct based on the question's
requirements.  Avoid any unnecessary analysis, assumptions, or
re-solving the problem.

Note:
- For intervals/ranges:  The student's answer must cover the EXACT
SAME range as the standard answer, NOT just any single value or
subset within that range;
- If the standard answer contains multiple solutions connected by
'or'/'and', all of them must be listed in the student's answer;
- If student's response does not mention any answer, it is
considered WRONG;
- You must be deterministic and rigorous - always declare the
answer as either CORRECT or WRONG;
- Small rounding differences are permitted if all the derivation
steps are correct.

Your response must include:
### Short Analysis
Provide a short and evidence-backed analysis between <analysis>
</analysis> tags, in which you should extract the final solution
value from the standard answer and the student's answer and judge
whether they are the same.

### Correctness
Based on the analysis, you should report a label CORRECT or WRONG
between <judge> </judge> tags (e.g., <judge>CORRECT</judge> or
<judge>WRONG</judge>).

### User Prompt
Problem:  {problem}

Standard Answer:  {standard_answer}

Student Answer:  {student_answer}
```

Table 3: LLM-as-a-judge prompt template for evaluating model responses

To ensure accurate scoring for evaluations in §3, we first validate all responses with `Math-Verify` (Kydlíček) and double check with `DeepSeek-V3` as a judge. We prompt DeepSeek-V3 for responses that are labeled as wrong by math-verify. Table 3 contains the exact prompt.

## C  BENCHMARK RESULTS

Here, we provide a detailed breakdown of all LLM performance on five math benchmarks and four coding benchmarks. The *Avg.* column is the same as the one in Table 1.

| Model | AIME 24 | AIME 25 | MATH-500 | Minerva | Olympiad | Avg. |
|---|---|---|---|---|---|---|
| *Low Benchmark Scores* | | | | | | |
| R1-Qwen-1.5B | 30.4 | 21.7 | 84.2 | 47.6 | 53.7 | 47.5 |
| R1-Llama-8B | 42.9 | 27.1 | 88.3 | 49.0 | 63.5 | 54.1 |
| DeepMath-1.5B | 37.5 | 29.2 | 90.1 | 54.8 | 62.6 | 54.8 |
| DeepScaleR-1.5B | 40.0 | 30.0 | 89.9 | 54.7 | 61.8 | 55.3 |
| OpenThinker3-1.5B | 52.1 | 39.6 | 92.2 | 43.7 | 68.4 | 59.2 |
| Qwen3-1.7B | 44.2 | 36.7 | 92.1 | 59.5 | 67.3 | 59.9 |
| *Medium Benchmark Scores* | | | | | | |
| R1-Qwen-7B | 55.4 | 38.3 | 94.3 | 64.3 | 70.8 | 64.6 |
| LIMO-32B | 55.8 | 41.7 | 95.4 | 70.5 | 73.0 | 67.3 |
| OpenThinker3-7B | 63.3 | 58.3 | 96.4 | 64.6 | 77.8 | 72.1 |
| R1-Qwen-32B | 67.9 | 52.1 | 95.4 | 69.9 | 76.5 | 72.3 |
| *High Benchmark Scores* | | | | | | |
| Qwen3-8B | 76.3 | 70.4 | 97.3 | 72.2 | 79.6 | 79.1 |
| QwQ-32B | 79.6 | 69.6 | 97.9 | 72.6 | 83.1 | 80.5 |
| Qwen3-32B | 78.3 | 71.7 | 97.5 | 75.0 | 82.3 | 81.0 |
| Qwen3-30B-A3B | 77.5 | 73.8 | 97.6 | 74.1 | 82.2 | 81.1 |
| AM-Thinking-32B | 80.4 | 77.9 | 98.4 | 72.8 | 83.5 | 82.6 |

Table 4: **Benchmark performance** (%) of 15 thinking LLMs. "Olympiad" stands for Olympiad-Bench and "Minerva" is the math subset in Minerva benchmark. "Avg" = unweighted mean of AIME 24, AIME 25, MATH-500, Minerva, and OlympiadBench.

| Model | CruxEval | HumanEval | HumanEval+ | MBPP | MBPP+ | Avg. |
|---|---|---|---|---|---|---|
| *Low Benchmark Scores* | | | | | | |
| R1-Qwen-1.5B | 35.4 | 66.7 | 60.7 | 48.5 | 55.2 | 53.3 |
| OpenThinker3-1.5B | 28.1 | 67.7 | 62.6 | 51.9 | 56.9 | 53.4 |
| DeepScaleR-1.5B | 41.5 | 74.2 | 66.3 | 53.1 | 59.0 | 58.8 |
| DeepMath-1.5B | 44.0 | 73.6 | 66.2 | 54.1 | 60.8 | 59.8 |
| *Medium Benchmark Scores* | | | | | | |
| Qwen3-1.7B | 75.2 | 83.3 | 75.2 | 68.5 | 71.2 | 74.7 |
| R1-Llama-8B | 73.3 | 85.4 | 79.7 | 70.8 | 70.9 | 76.0 |
| R1-Qwen-7B | 73.8 | 88.3 | 82.3 | 71.7 | 72.4 | 77.7 |
| OpenThinker-7B | 83.6 | 84.7 | 75.9 | 80.4 | 75.9 | 80.1 |
| *High Benchmark Scores* | | | | | | |
| Qwen3-8B | 91.2 | 91.5 | 85.3 | 88.2 | 81.0 | 87.4 |
| Qwen3-32B | 94.9 | 95.3 | 89.0 | 84.0 | 75.9 | 87.8 |
| Qwen3-30B-A3B | 93.8 | 94.4 | 86.2 | 90.4 | 80.3 | 89.0 |
| R1-Qwen-32B | 92.0 | 96.3 | 88.9 | 90.0 | 81.4 | 89.7 |
| AM-Thinking-32B | 94.3 | 94.4 | 88.0 | 91.6 | 80.4 | 89.7 |
| QwQ-32B | 93.8 | 96.5 | 88.0 | 93.3 | 80.7 | 90.5 |

Table 5: **Benchmark performance** (%) of 14 thinking LLMs. "MBPP+" and "HumanEval+" are two synthesized splits from EvalPlus (Liu et al., 2023). "Avg" = unweighted mean of CruxEval, MBPP, MBPP-Plus, HumanEval, and HumanEval-Plus.

## D  RECOVERABILITY TEST

Table 6 reports a breakdown of model recoverability performance on shared subset across different positions (%) of the original trajectories. Table 7 reports the results of ablation study explained in §3.2, where the first paragraph of model reasoning is preserved. The subscripts in Table 7 equal the difference between the corresponding values in Table 7 and Table 6, showing the changes in recoverability induced by preserving the original opening.

| Model | 0% | 20% | 40% | 60% | 80% | Avg. | Benchmark Avg. |
|---|---|---|---|---|---|---|---|
| R1-Qwen-1.5B | 44.0 | 66.0 | 64.0 | 67.0 | 62.0 | 60.6 | 47.5 |
| R1-Llama-8B | 65.5 | 81.5 | 84.5 | 82.5 | 93.0 | 81.4 | 54.1 |
| DeepMath-1.5B | 71.5 | 94.0 | 90.0 | 94.0 | 90.5 | 88.0 | 54.8 |
| DeepScaleR-1.5B | 61.5 | 88.0 | 89.5 | 85.0 | 88.0 | 82.4 | 55.3 |
| OpenThinker3-1.5B | 89.0 | 95.5 | 96.5 | 98.0 | 97.0 | 95.2 | 59.2 |
| Qwen3-1.7B | 97.0 | 99.5 | 99.0 | 98.5 | 98.0 | 98.4 | 59.9 |
| R1-Qwen-7B | 48.5 | 77.0 | 79.0 | 82.5 | 80.5 | 73.5 | 64.6 |
| LIMO-32B | 18.0 | 29.0 | 36.0 | 32.5 | 31.0 | 29.3 | 67.3 |
| OpenThinker3-7B | 81.5 | 87.0 | 89.0 | 84.5 | 86.0 | 85.6 | 72.1 |
| R1-Qwen-32B | 21.0 | 70.5 | 78.5 | 90.5 | 88.5 | 69.8 | 72.3 |
| Qwen3-8B | 71.0 | 88.5 | 89.0 | 91.5 | 89.5 | 85.9 | 79.1 |
| QwQ-32B | 53.0 | 79.5 | 86.5 | 88.5 | 91.0 | 79.7 | 80.5 |
| Qwen3-32B | 32.5 | 74.5 | 88.5 | 81.0 | 82.5 | 71.8 | 81.0 |
| Qwen3-30B-A3B | 68.0 | 90.5 | 93.5 | 91.5 | 95.5 | 87.8 | 81.1 |
| AM-Thinking-32B | 16.5 | 29.0 | 36.5 | 41.0 | 44.0 | 33.4 | 82.6 |

Table 6: **Recoverability (shared)** results (on 200 questions fully solved by all 15 LLMs eight out of eight). 0%, 20%, 40%, 60%, 80% are the positions of original reasoning where distraction is introduced. "Avg." column averages across all the positions. "Benchmark Avg." is from Table 4

| Model | 0% | 20% | 40% | 60% | 80% | Avg. | Benchmark Avg. |
|---|---|---|---|---|---|---|---|
| R1-Qwen-1.5B | $89.0_{+45.0}$ | $94.0_{+28.0}$ | $91.0_{+27.0}$ | $89.5_{+22.5}$ | $84.0_{+22.0}$ | $89.5_{+28.9}$ | 47.5 |
| R1-Llama-8B | $95.5_{+30.0}$ | $96.5_{+15.0}$ | $97.0_{+12.5}$ | $91.5_{+9.0}$ | $87.0_{-6.0}$ | $93.5_{+12.1}$ | 54.1 |
| DeepMath-1.5B | $99.0_{+27.5}$ | $98.5_{+4.5}$ | $98.5_{+8.5}$ | $98.0_{+4.0}$ | $95.0_{+4.5}$ | $97.8_{+9.8}$ | 54.8 |
| DeepScaleR-1.5B | $97.0_{+35.5}$ | $97.5_{+9.5}$ | $97.5_{+8.0}$ | $98.0_{+13.0}$ | $86.0_{-2.0}$ | $95.2_{+12.8}$ | 55.3 |
| OpenThinker3 1.5B | $96.5_{+7.5}$ | $98.0_{+2.5}$ | $97.0_{+0.5}$ | $100.0_{+2.0}$ | $96.0_{-1.0}$ | $97.5_{+2.3}$ | 59.2 |
| Qwen3-1.7B | $100.0_{+3.0}$ | $100.0_{+0.5}$ | $100.0_{+1.0}$ | $100.0_{+1.5}$ | $82.0_{-16.0}$ | $96.4_{-2.0}$ | 59.9 |
| R1-Qwen-7B | $91.5_{+43.0}$ | $95.5_{+18.5}$ | $91.0_{+12.0}$ | $89.5_{+7.0}$ | $85.0_{+4.5}$ | $90.5_{+17.0}$ | 64.6 |
| LIMO-32B | $58.0_{+40.0}$ | $57.5_{+28.5}$ | $54.5_{+18.5}$ | $60.5_{+28.0}$ | $53.5_{+22.5}$ | $56.8_{+27.5}$ | 67.3 |
| OpenThinker3-7B | $93.0_{+11.5}$ | $94.5_{+7.5}$ | $96.0_{+7.0}$ | $96.5_{+12.0}$ | $85.0_{-1.0}$ | $93.0_{+7.4}$ | 72.1 |
| R1-Qwen-32B | $74.5_{+53.5}$ | $80.5_{+10.0}$ | $90.0_{+11.5}$ | $93.5_{+3.0}$ | $85.0_{-3.5}$ | $84.7_{+14.9}$ | 72.3 |
| Qwen3-8B | $95.5_{+24.5}$ | $97.0_{+8.5}$ | $97.5_{+8.5}$ | $97.0_{+5.5}$ | $80.0_{-9.5}$ | $93.4_{+7.5}$ | 79.1 |
| QwQ-32B | $64.5_{+11.5}$ | $73.0_{-6.5}$ | $81.0_{-5.5}$ | $90.0_{+1.5}$ | $86.5_{-4.5}$ | $79.0_{-0.7}$ | 80.5 |
| Qwen3-32B | $75.0_{+42.5}$ | $87.0_{+12.5}$ | $95.5_{+7.0}$ | $92.5_{+11.5}$ | $67.5_{-15.0}$ | $83.5_{+11.7}$ | 81.0 |
| Qwen3-30B-A3B | $83.5_{+15.5}$ | $88.0_{-2.5}$ | $91.0_{-2.5}$ | $94.0_{+2.5}$ | $66.0_{-29.5}$ | $84.5_{-3.3}$ | 81.1 |
| AM-Thinking-32B | $55.0_{+38.5}$ | $53.0_{+24.0}$ | $60.0_{+23.5}$ | $75.0_{+34.0}$ | $42.5_{-1.5}$ | $57.1_{+23.7}$ | 82.6 |

Table 7: Ablation Study: **Recoverability (shared)** results with original beginning (on 200 questions fully solved by all 15 LLMs eight out of eight). 0%, 20%, 40%, 60%, 80% are the positions of original reasoning where distraction is introduced. "Avg." averages across all the positions. "Benchmark Avg." is from Table 4

Table 8 and Table 10 report detailed breakdown of recoverability on individual subset; the former sets the length of distracting steer $r^{\text{steer}}$ to be 0.2 times of the reasoning trajectory by default, whereas the latter sets to 0.4 of the reasoning trajectory.

| Model | 0% | 20% | 40% | 60% | 80% | Avg. | Benchmark Avg. |
|---|---|---|---|---|---|---|---|
| R1-Qwen-1.5B | 24.0 | 40.8 | 40.8 | 38.8 | 48.4 | 38.6 | 47.5 |
| R1-Llama-8B | 32.0 | 38.4 | 49.2 | 57.6 | 79.8 | 49.6 | 54.1 |
| DeepMath-1.5B | 54.4 | 61.6 | 61.6 | 64.0 | 67.6 | 61.8 | 54.8 |
| DeepScaleR-1.5B | 35.2 | 54.0 | 56.8 | 57.6 | 60.8 | 52.9 | 55.3 |
| OpenThinker3-1.5B | 58.0 | 69.6 | 77.6 | 76.0 | 78.0 | 71.8 | 59.2 |
| Qwen3-1.7B | 58.4 | 70.4 | 74.4 | 85.2 | 84.4 | 74.6 | 59.9 |
| R1-Qwen-7B | 38.4 | 48.0 | 46.4 | 50.4 | 45.6 | 45.8 | 64.6 |
| LIMO-32B | 8.8 | 21.2 | 18.8 | 20.0 | 23.6 | 18.5 | 67.3 |
| OpenThinker3-7B | 63.2 | 72.4 | 76.4 | 77.6 | 82.8 | 74.5 | 72.1 |
| R1-Qwen-32B | 8.4 | 37.6 | 53.6 | 58.0 | 70.4 | 45.6 | 72.3 |
| Qwen3-8B | 51.6 | 64.4 | 73.2 | 76.0 | 78.8 | 68.8 | 79.1 |
| QwQ-32B | 50.0 | 54.5 | 64.8 | 68.8 | 74.8 | 62.6 | 80.5 |
| Qwen3-32B | 23.6 | 53.6 | 67.2 | 66.4 | 73.6 | 56.9 | 81.0 |
| Qwen3-30B-A3B | 36.8 | 61.6 | 68.8 | 67.6 | 65.2 | 60.0 | 81.1 |
| AM-Thinking-32B | 19.6 | 26.8 | 29.6 | 26.4 | 24.0 | 25.3 | 82.6 |

Table 8: **Recoverability (individual)** results (on 200 randomly sampled questions for each of 15 LLMs). We sample questions according to the inverse proportions of solve rates. 0%, 20%, 40%, 60%, 80% are the positions of original reasoning where distraction is introduced. "Avg." averages across all the positions. "Benchmark Avg." is from Table 4

| Model | 1/8 | 2/8 | 3/8 | 4/8 | 5/8 | 6/8 | 7/8 | 8/8 |
|---|---|---|---|---|---|---|---|---|
| R1-Qwen-1.5B | 16.1 | 27.7 | 28.6 | 33.8 | 36.5 | 40.0 | 48.1 | 58.9 |
| R1-Llama-8B | 30.5 | 23.3 | 39.4 | 51.8 | 56.4 | 47.9 | 65.2 | 70.0 |
| DeepMath-1.5B | 26.2 | 45.6 | 40.1 | 56.8 | 51.3 | 55.5 | 81.1 | 80.7 |
| DeepScaleR-1.5B | 21.9 | 30.5 | 32.6 | 35.3 | 53.9 | 61.5 | 62.7 | 73.3 |
| OpenThinker-1.5B | 51.4 | 51.2 | 56.8 | 56.5 | 70.0 | 71.9 | 80.5 | 93.0 |
| Qwen3-1.7B | 40.9 | 38.0 | 52.3 | 65.2 | 70.6 | 80.6 | 84.7 | 91.3 |
| R1-Qwen-7B | 28.4 | 32.0 | 34.7 | 46.4 | 47.7 | 45.0 | 44.9 | 56.9 |
| LIMO-32B | 15.3 | 22.1 | 21.2 | 20.0 | 14.3 | 13.3 | 28.7 | 15.4 |
| OpenThinker-7B | 56.9 | 61.5 | 52.0 | 69.2 | 56.7 | 70.8 | 74.5 | 83.7 |
| R1-Qwen-32B | 24.7 | 35.4 | 40.0 | 40.0 | 32.5 | 41.9 | 48.0 | 52.9 |
| Qwen3-8B | 45.0 | 34.3 | 49.2 | 55.0 | 46.7 | 63.6 | 71.2 | 77.9 |
| QwQ-32B | 35.0 | 49.1 | 25.0 | 33.3 | 55.7 | 58.8 | 59.2 | 69.4 |
| Qwen3-32B | 37.3 | 43.3 | 32.5 | 42.9 | 44.6 | 56.4 | 45.3 | 64.7 |
| Qwen3-30B-A3B | 42.9 | 25.0 | 38.7 | 54.5 | 55.0 | 60.0 | 60.0 | 65.9 |
| AM-Thinking-32B | 11.1 | 22.2 | 27.5 | 12.5 | 23.6 | 16.7 | 21.4 | 28.9 |

Table 9: **Recoverability (individual)** results (on 200 randomly sampled questions in total for each of 15 LLMs) across different solve rates (N=8). Across all LLMs, we observe that recoverability is positively correlated with solve rates, since models need to solve the original question, even after successfully backtracking from distraction.

| Model | 0% | 20% | 40% | 60% | Avg. | Benchmark Avg. |
|---|---|---|---|---|---|---|
| R1-Qwen-1.5B | 11.6 | 26.0 | 27.6 | 24.0 | 22.3 | 47.5 |
| R1-Llama-8B | 29.2 | 43.2 | 54.8 | 56.4 | 45.9 | 54.1 |
| DeepMath-1.5B | 38.8 | 54.0 | 43.6 | 51.2 | 46.9 | 54.8 |
| DeepScaleR-1.5B | 24.8 | 50.0 | 53.2 | 50.4 | 44.6 | 55.3 |
| OpenThinker3-1.5B | 52.4 | 70.8 | 68.8 | 78.8 | 67.7 | 59.2 |
| Qwen3-1.7B | 59.2 | 73.2 | 76.4 | 81.2 | 72.5 | 59.9 |
| R1-Qwen-7B | 25.6 | 41.2 | 39.2 | 36.4 | 35.6 | 64.6 |
| LIMO-32B | 6.0 | 10.8 | 16.8 | 17.6 | 12.8 | 67.3 |
| OpenThinker3-7B | 59.6 | 72.0 | 70.0 | 73.2 | 68.7 | 72.1 |
| R1-Qwen-32B | 10.8 | 36.8 | 49.2 | 62.0 | 39.7 | 72.3 |
| Qwen3-8B | 50.4 | 67.2 | 71.2 | 76.0 | 66.2 | 79.1 |
| QwQ-32B | 44.8 | 52.0 | 61.2 | 68.4 | 56.6 | 80.5 |
| Qwen3-32B | 23.2 | 59.6 | 62.4 | 65.6 | 52.7 | 81.0 |
| Qwen3-30B-A3B | 31.6 | 53.2 | 62.0 | 59.6 | 51.6 | 81.1 |
| AM-Thinking-32B | 22.8 | 33.6 | 29.6 | 26.0 | 28.0 | 82.6 |

Table 10: **Recoverability (individual)** results with **40% of distracting reasoning**. We control length of distraction to be 40% of distracting reasoning trace (default 20% in Table 6). The sampled questions are the same as in Table 6. 0%, 20%, 40%, 60% are the positions of original reasoning where distraction is injected. "Avg." averages across all positions. "Benchmark Avg." is from Table 4

# E  ADDITIONAL ABLATION STUDIES

## E.1  DISTRACTOR DESIGN RATIONALE AND CROSS-MODEL ABLATION

**Stitching procedure.** When constructing the steered trajectory $[r^{\text{og}}, r^{\text{steer}}]$, we truncate $r^{\text{og}}$ at the end of the nearest sentence to preserve coherence. The distracting steer $r^{\text{steer}}$ is then appended on a new line, prefixed with a short transition phrase ("Wait. Let me think") that mimics the self-correction style common in reasoning traces (see Figure 2). The entire steered trajectory is placed directly inside the model's assistant-side thinking (e.g., within <think> tags).

**Why sample from the same model on a different question?** Our recoverability protocol constructs distractors by sampling from the evaluated model $M$ itself, conditioned on a different question $q'$

(§2.1). This design has three advantages. First, sampling from $q'$ **guarantees** that blindly completing the distractor leads to an incorrect answer for the original question $q$, without requiring an external LLM judge to verify incorrectness. Second, the approach is **model-agnostic**: it does not assume knowledge of which collaborator models will be encountered at deployment time. If specific collaborators are known a priori, our protocol generalizes straightforwardly to sampling distractors from those models. Third, the strategy is **scalable**, requiring only one additional generation per test point.

This design is motivated by prior findings that incomplete reasoning traces can mislead LLMs into naively continuing them rather than backtracking. Yang et al. (2025b) inject four types of unhelpful thoughts (uninformative, irrelevant, misdirecting, incorrect) into reasoning traces and find that models tend to follow the injected reasoning. Similarly, Zhou et al. (2024) show that irrelevant thoughts in chain-of-thought prompting degrade accuracy by 1.4–19.8%, confirming that LLMs are susceptible to off-distribution reasoning content.

**Cross-model ablation.** To verify that same-model sampling does not introduce systematic bias, we conduct an ablation where distractors are instead sampled from five randomly selected models. As shown in Table 11, recoverability scores remain consistent across both settings, confirming that our protocol is robust to the choice of distractor source.

| Model | Recoverability (Sh.) | |
| --- | --- | --- |
| | Same Model | Different Model |
| R1-Qwen-1.5B | 60.6 | 68.8 |
| DeepScaleR-1.5B | 82.4 | 80.7 |
| R1-Llama-8B | 81.4 | 79.9 |
| DeepMath-1.5B | 88.0 | 92.7 |
| OpenThinker3-1.5B | 95.2 | 91.7 |
| Qwen3-1.7B | 98.4 | 97.5 |
| R1-Qwen-7B | 73.5 | 70.0 |
| LIMO-32B | 29.3 | 25.2 |
| OpenThinker3-7B | 85.6 | 85.5 |
| R1-Qwen-32B | 69.8 | 66.1 |
| Qwen3-8B | 85.9 | 85.6 |
| QwQ-32B | 79.7 | 77.8 |
| Qwen3-32B | 71.8 | 73.8 |
| Qwen3-30B-A3B | 87.8 | 86.7 |
| AM-Thinking-32B | 33.4 | 33.9 |

Table 11: **Recoverability under different distractor sources.** "Same Model": distractors sampled from the evaluated model (corresponds to Recoverability (Sh.)" in Table 1). "Different Model": distractors sampled from five other randomly selected models.

### E.2 GUIDABILITY RESULTS ON ORIGINALLY SOLVABLE PROBLEMS

Section 3.2 shows that LLMs struggle to leverage partial thinking traces from a guidance model on questions beyond their capabilities (solve rate $\leq 1/8$). To rule out distribution shift between the guidance model and the evaluated model as a confounding factor, we test guidability on problems that models can fully solve independently (solve rate $= 8/8$). As shown in Table 12, guidability reaches nearly $100\%$ across all models on this subset, confirming that the observed limitations stem from models' inherent capabilities rather than cross-model distribution mismatch.

| Model | Guidability (on fully solvable set) |
|---|---|
| R1-Qwen-1.5B | 98.5 |
| DeepScaleR-1.5B | 98.6 |
| R1-Llama-8B | 98.4 |
| DeepMath-1.5B | 98.5 |
| OpenThinker3-1.5B | 99.7 |
| Qwen3-1.7B | 98.9 |
| R1-Qwen-7B | 99.0 |
| LIMO-32B | 98.1 |
| OpenThinker3-7B | 98.9 |
| R1-Qwen-32B | 98.3 |
| Qwen3-8B | 98.7 |
| QwQ-32B | 98.8 |
| Qwen3-32B | 98.7 |
| Qwen3-30B-A3B | 98.4 |
| AM-Thinking-32B | 98.1 |

Table 12: **Guidability on originally solvable problems.** Results for problems that models solve independently (solve rate $= 8/8$).

### E.3 CODING GUIDABILITY

Table 17 reports the detailed breakdown of coding guidability (individual) results across seven models and four steer proportions. The coding guidability scores are substantially higher than their math counterparts (Table 14). To assess whether models are genuinely continuing reasoning from the steer or simply benefiting from information already present in it, we conduct an answer-forcing test: we append a closing `</think>` tag immediately after the steer, forcing the model to produce an answer without any additional reasoning. The resulting pass rates are reported as subscripts. At higher steer proportions, the answer-forcing rate meets or exceeds the guidability score for most models, indicating that models cannot genuinely continue reasoning from the steer—and in some cases actively hurt performance by overriding correct information already present. For DeepScaleR-1.5B and DeepMath-1.5B, answer-forcing matches guidability across all ratios.

## F GUIDABILITY TEST

Table 13 reports the number of unique problems and guiding trajectories used per guiding model (sub-column) for each LLM (row). Table 14 reports guidability (individual) results for different length of the guiding steers measured by $x\%$ of the trajectories. Similarly, Table 15 reports breakdown of guidability on shared subset. Table 16 groups guidability (individual) scores by the guiding models (column) for each LLM (row)

| | # of Problems | | | # of Trajectories | | |
|---|---|---|---|---|---|---|
| | **DeepSeek-R1** | **Qwen-3** | **QwQ-32B** | **DeepSeek-R1** | **Qwen-3** | **QwQ-32B** |
| DeepMath-1.5B | 152 | 198 | 302 | 231 | 268 | 302 |
| DeepScaleR-1.5B | 154 | 196 | 311 | 234 | 269 | 311 |
| LIMO-Qwen-32B | 100 | 137 | 185 | 142 | 172 | 185 |
| OpenThinker3-1.5B | 151 | 199 | 270 | 236 | 278 | 270 |
| OpenThinker3-7B | 101 | 146 | 163 | 146 | 186 | 163 |
| Qwen3-1.7B | 130 | 175 | 245 | 192 | 233 | 245 |
| R1-Llama-8B | 151 | 196 | 266 | 229 | 269 | 266 |
| R1-Qwen-1.5B | 168 | 213 | 363 | 261 | 290 | 363 |
| R1-Qwen-7B | 107 | 156 | 190 | 151 | 195 | 190 |
| R1-Qwen-32B | 94 | 145 | 162 | 134 | 182 | 162 |

Table 13: **Guidability statistics**: unique number of problems and trajectories per guiding model (column) for different student models (row) for **Guidability (individual)** test.

| Model | 20% | 40% | 60% | 80% | Avg | Benchmark Avg. |
|---|---|---|---|---|---|---|
| R1-Qwen-1.5B | $14.6_{7.7}$ | $23.1_{17.2}$ | $33.2_{31.3}$ | $43.0_{46.2}$ | $28.4_{25.6}$ | 47.5 |
| R1-Llama-8B | $20.8_{5.4}$ | $29.6_{15.7}$ | $40.0_{27.6}$ | $49.7_{34.8}$ | $35.0_{21.8}$ | 54.1 |
| DeepMath-1.5B | $13.6_{7.2}$ | $21.1_{16.2}$ | $31.2_{27.5}$ | $42.3_{40.6}$ | $27.1_{22.9}$ | 54.8 |
| DeepScaleR-1.5B | $15.7_{7.5}$ | $23.2_{15.7}$ | $34.6_{28.1}$ | $45.6_{41.8}$ | $29.8_{23.3}$ | 55.3 |
| OpenThinker3-1.5B | $18.1_{11.0}$ | $30.6_{21.4}$ | $36.1_{32.3}$ | $46.0_{42.3}$ | $32.7_{26.9}$ | 59.2 |
| Qwen3-1.7B | $18.2_{5.8}$ | $23.7_{11.8}$ | $34.8_{20.6}$ | $42.8_{33.8}$ | $29.9_{18.0}$ | 59.9 |
| R1-Qwen-7B | $10.8_{3.5}$ | $16.2_{6.3}$ | $22.0_{13.1}$ | $29.9_{25.4}$ | $19.7_{12.1}$ | 64.6 |
| LIMO-32B | $12.6_{2.6}$ | $18.8_{4.8}$ | $24.4_{11.6}$ | $30.0_{21.8}$ | $21.5_{10.2}$ | 67.3 |
| OpenThinker3-7B | $11.1_{6.5}$ | $20.0_{10.1}$ | $22.6_{15.4}$ | $28.7_{23.4}$ | $20.6_{13.8}$ | 72.1 |
| R1-Qwen-32B | $14.2_{3.8}$ | $19.7_{6.1}$ | $24.9_{12.4}$ | $31.2_{22.6}$ | $22.5_{11.2}$ | 72.3 |

Table 14: **Guidability (individual)** results (on all questions with solve rate $\leq \frac{1}{8}$ **for each individual model**). 20%, 40%, 60%, 80% are proportion of teacher reasoning revealed to the student model in its thinking window. The subscript value is the percentage of cases where teachers **have derived the solution**. "Avg" is the average across different proportions. "Benchmark Avg" is the same as in Table 4.

| Model | 20% | 40% | 60% | 80% | Avg | Benchmark Avg. |
|---|---|---|---|---|---|---|
| R1-Qwen-1.5B | 1.2 | 0.9 | 4.1 | 5.8 | 3.0 | 47.5 |
| R1-Llama-8B | 5.2 | 5.8 | 10.4 | 13.3 | 8.7 | 54.1 |
| DeepMath-1.5B | 0.9 | 0.9 | 4.6 | 7.2 | 3.4 | 54.8 |
| DeepScaleR-1.5B | 1.2 | 0.9 | 5.2 | 9.0 | 4.1 | 55.3 |
| OpenThinker3-1.5B | 1.7 | 5.5 | 7.0 | 8.4 | 5.7 | 59.2 |
| Qwen3-1.7B | 2.3 | 3.2 | 7.8 | 11.0 | 6.1 | 59.9 |
| R1-Qwen-7B | 2.6 | 5.2 | 6.4 | 9.9 | 6.0 | 64.6 |
| LIMO-32B | 4.9 | 7.5 | 10.1 | 12.8 | 8.8 | 67.3 |
| OpenThinker3-7B | 4.9 | 9.0 | 9.6 | 12.8 | 9.1 | 72.1 |
| R1-Qwen-32B | 4.1 | 7.5 | 11.0 | 14.2 | 9.2 | 72.3 |

Table 15: **Guidability (shared)** results (on questions with solve rate $\leq \frac{1}{8}$ **across all ten models**). 20%, 40%, 60%, 80% are proportion of teacher reasoning revealed to the student model in its thinking window. "Avg" is the average across different proportions. "Benchmark Avg" is the same as in Table 4.

| Model | DeepSeek-R1 | QwQ-32B | Qwen3-235B-A22B | Benchmark Avg. |
|---|---|---|---|---|
| R1-Qwen-1.5B | 28.2 | 30.4 | 26.2 | 47.5 |
| DeepMath-1.5B | 29.0 | 26.2 | 26.3 | 54.8 |
| DeepScaleR-1.5B | 30.9 | 31.1 | 27.3 | 55.3 |
| R1-Llama-8B | 37.8 | 34.4 | 33.2 | 54.1 |
| Qwen3-1.7B | 33.4 | 31.1 | 25.6 | 59.9 |
| OpenThinker3-1.5B | 35.7 | 30.6 | 32.3 | 59.2 |
| R1-Qwen-7B | 22.0 | 19.6 | 18.7 | 64.6 |
| LIMO-32B | 24.5 | 24.6 | 15.7 | 67.3 |
| R1-Qwen-32B | 23.5 | 23.0 | 21.9 | 72.3 |
| OpenThinker3-7B | 22.9 | 21.4 | 18.0 | 77.8 |

Table 16: **Guidability (individual)** results (teacher model comparison). Each teacher model averages across **Guidability (individual)** scores for all proportions, 20%, 40%, 60%, 80%, in Table 14

| Model | 20% | 40% | 60% | 80% | Avg | Benchmark Avg. |
|---|---|---|---|---|---|---|
| R1-Qwen-1.5B | $19.6_{18.0}$ | $32.8_{29.5}$ | $47.0_{45.0}$ | $60.8_{66.0}$ | $40.1_{39.6}$ | 53.3 |
| R1-Llama-8B | $27.4_{17.5}$ | $46.2_{30.7}$ | $63.4_{54.1}$ | $80.5_{75.1}$ | $54.4_{44.3}$ | 76.0 |
| DeepMath-1.5B | $17.4_{14.5}$ | $30.2_{27.0}$ | $42.0_{41.9}$ | $55.3_{60.0}$ | $36.2_{35.9}$ | 59.8 |
| DeepScaleR-1.5B | $16.9_{16.0}$ | $28.8_{27.7}$ | $42.7_{42.2}$ | $60.7_{60.8}$ | $37.3_{36.7}$ | 58.8 |
| OpenThinker3-1.5B | $29.4_{10.4}$ | $43.2_{23.0}$ | $54.9_{40.6}$ | $70.2_{66.6}$ | $49.4_{35.1}$ | 53.4 |
| Qwen3-1.7B | $26.9_{16.8}$ | $48.9_{34.0}$ | $67.3_{58.4}$ | $82.6_{77.4}$ | $56.4_{46.7}$ | 74.7 |
| R1-Qwen-7B | $27.0_{18.9}$ | $47.1_{35.7}$ | $61.8_{55.1}$ | $76.4_{75.3}$ | $53.1_{46.2}$ | 77.7 |

Table 17: **Coding Guidability (individual)** results (on all coding questions with solve rate $\leq \frac{1}{8}$ **for each individual model**). 20%, 40%, 60%, 80% are proportion of teacher reasoning revealed to the student model. The subscript value is the answer-forcing pass rate: the percentage of cases where the model produces the correct answer when forced to skip reasoning after receiving the steer (§E.3). "Avg" averages across proportions. "Benchmark Avg" is from Table 4.

## G  ADDITIONAL CONTROL STUDY

**Supervised Fine-Tuning Hyperparameters.** We perform full fine-tuning on `Qwen2.5-1.5B` and `Qwen2.5-3B` base models for 5 epochs. The max tokens is set to 16K, batch size 64, learning rate 2e-5, warmup ratio 0.1, max gradient norm 1.0, weight decay 0.01.

**Ablation Study.** We compare the effects of distillation teachers on `Qwen2.5-7B` models. We observe similar patterns as discussed in §4.1, where AM-Distill models achieve worse recoverability compared to QwQ-/Qwen3-Distill models. The guidability scores are not measured since the benchmark performance are too high to collect sufficient qualified problems.

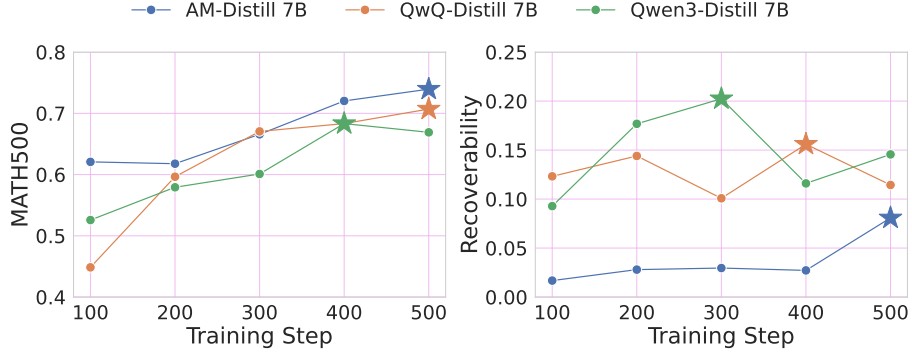

Figure 8: Qwen2.5 7B models distilled from AM (Thinking-v1) 32B also shows lower recoverability than those distilled from QwQ 32B or Qwen 32B, while having similar benchmark performance; the gap is **significant for all steps** (p $\leq$ 0.005). Stars mark each model's peak over training steps.

