# OpenReview forum: "Off-Trajectory Reasoning: Can LLMs Collaborate on Reasoning Trajectories?"
_ICLR.cc/2026/Conference — ICLR 2026 Poster_

### Official Review · Reviewer_BJqx · 2025-10-25

**Soundness:** 2
**Presentation:** 2
**Contribution:** 2
**Rating:** 6
**Confidence:** 3

**Summary:**

1. It investigates whether large language models (LLMs) trained for solo reasoning—i.e., generating complete reasoning chains independently—are capable of effective collaborative reasoning, where they must build on or recover from partial reasoning traces produced by other agents (e.g., other models or humans).
2. Stronger solo-reasoning models are often more fragile under distraction—high-benchmark models like AM-Thinking-32B show poor recoverability.
3. Reinforcement Learning (RL) after supervised fine-tuning (SFT) significantly improves both recoverability and guidability.

**Strengths:**

1. This work is among the first to systematically probe LLMs’ robustness and adaptability in multi-agent reasoning contexts.
2. The twin-test framework—Recoverability and Guidability—is conceptually original. It cleanly isolates two extremes of collaborative interaction (resisting distraction vs. leveraging guidance) and operationalizes them in a reproducible, controlled manner.
3. The stronger solo reasoners are more fragile under distraction, and no model effectively uses correct guidance to exceed its solo capability.

**Weaknesses:**

1. It simulates collaboration via a static, one-shot injection of an off-trajectory steer (either distracting or guiding), followed by continuation from the main model. This simplification overlooks more realistic collaborative dynamics—e.g., iterative back-and-forth exchanges, multi-turn corrections, or mixed-quality steers (partially correct, partially wrong).
2. All experiments are conducted on mathematical reasoning benchmarks. While math is a strong testbed for structured reasoning, it may not generalize to other domains (e.g., commonsense, legal, or scientific reasoning) where reasoning trajectories are less formalized or more ambiguous.

**Questions:**

1. Since the models used in exp are trained with different training recipe, could you use open-sourced datasets and the numbers of training dataset to study the Recoverability and Guidability at the SFT-stage?
2. The models listed in exp are almost trained with sft-loss, could you use open-sourced datasets and RL-training to study the impact of RL for Recoverability and Guidability?

---

> ### Author Response · Authors · 2025-11-24
> **Official Response to Review BJqx Part 1**
>
> **[Response to Weakness 1]**
>
> First of all, we want to emphasize the fact that our test protocol and evaluation results, to our best knowledge, is the first work in this field to systematically evaluate current off-the-shelf reasoning model collaborations. This work should be highlighted by its novelty and originality.
>
> Second, our framework can be easily extended to include distractors or guides at multiple steps instead of just once. The framework’s design is not limited to one-shot injection and can be naturally extended to multi-turn settings; only that we study a particular instantiation with one-shot injections. We do this since our one-shot injection test already exposes the critical limitations in current LLMs’ multi-model collaborative capabilities, rendering multi-turn interaction study unnecessary at this stage.
>
> Our point is that, while any multi-turn/ multi-model collaboration can be much richer and complex, e.g., “iterative back-and-forth exchanges, multi-turn corrections, or mix-quality steers.”, such a framework is unnecessary to study the current model artifacts. As with any benchmark, we expect that as multi-turn collaboration capabilities improve, we will need newer benchmarks that capture richer interactions as the reviewer suggests. However, our twin tests suffice for the current off-the-shelf models, as it already reveals the lack of robustness in reasoning and and limitations of integrating teacher models’ guidance beyond their capabilities.
>
> **[Response to Weakness 2]**
>
> In our work, we deliberately focused on math benchmarks for conducting twin tests as the majority of recent reasoning work and artifacts in open-source community (DeepMath, DeepScaleR, LIMO-32B) have exclusively trained on math data. This focus is advantageous because (a) there exist more open-weight model artifacts in math domain for us to study off-trajectory reasoning (e.g., LIMO-32B frequently fails to generate CoT thinking for non-math questions); (b) many prior works in LLM reasoning (Liu Zichen et al., 2025; Dapo et al., 2025; Liu Zihan et al., 2025; Zelikman, 2022; Zelikman, 2024 ) develop and validate their proposed methods or training recipes primarily on math tasks.  In addition, prior work (e.g., Sprague et al.) shows that science benchmarks (e.g., GPQA, MMLU) benefit much less from long CoT, as these tasks are predominantly knowledge-based, and even when reasoning is involved, it is often mathematical in nature.
>
> Nevertheless, we did conduct additional experiments in coding. We run our Recoverability and Guidability twin tests on four benchmarks (MBPP, HumanEval, CruxEval, EvalPlus). The results are reported in the table below. We find that the findings from our paper in the math domain hold on the code domain as well: (1) models with stronger solo reasoning do not consistently achieve higher recoverability, and (2) guidability is not strictly correlated with benchmark strength. Interesting, the AM-Thinking-32B, which we specially highlighted as a model with a strong benchmark scores and very poor recoverability on math reports a similar gap between on the code domain as well. This highlights an inherent weakness of this model that is not reflected in standard benchmarks but is exposed via our twin tests. Since our central findings transfer to coding tasks, we believe this addresses the generalization concern. For other domains, many open-source models (e.g., DeepScaleR-1.5B, DeepMath-1.5B, OpenThinker3-1.5B/1.7B, LIMO-32B, AM-Thinking-32B) are exclusively optimized for math and/or code, and we find that they often fail to produce coherent CoT reasoning on non-math tasks, making evaluation unreliable. In addition, these domains (e.g., legal reasoning) often lack high-quality reasoning data, preventing us from conducting the corresponding control studies in Section 4.

---

> > ### Author Response · Authors · 2025-11-24
> >
> > | Model              | Benchmark Avg. | Recoverability (Sh.) | Recoverability (Ind.) | Guidability (Sh.) | Guidability (Ind.) |
> > |-------------------|----------------|-----------------------|------------------------|--------------------|---------------------|
> > | R1-Qwen-1.5B       | 53.3          | 23.0 (+0)            | 17.9 (+0)             | 28.0 (+2)         | 52.8 (+2)          |
> > | OpenThinker-1.5B   | 53.4          | 59.2 (+10)           | 47.6 (+10)            | 37.3 (+2)         | 53.4 (+2)          |
> > | DeepScaleR-1.5B    | 58.8          | 45.9 (+4)            | 34.3 (+2)             | 27.1 (-2)         | 42.7 (-1)          |
> > | DeepMath-1.5B      | 59.8          | 53.8 (+6)            | 40.6 (+4)             | 27.4 (-2)         | 42.2 (-3)          |
> > | Qwen3-1.7B         | 74.7          | 70.2 (+9)            | 52.1 (+8)             | 44.4 (+0)         | 56.6 (+1)          |
> > | R1-Llama-8B        | 76.0          | 50.1 (+3)            | 43.7 (+5)             | 48.2 (+1)         | 57.1 (+1)          |
> > | R1-Qwen-7B         | 77.7          | 35.2 (-5)            | 27.5 (-5)             | 47.9 (-1)         | 55.7 (-2)          |
> > | OpenThinker-7B     | 80.1          | 68.4 (+5)            | 61.6 (+6)             | NA                | NA                 |
> > | Qwen3-8B           | 87.4          | 54.4 (+2)            | 42.2 (+0)             | NA                | NA                 |
> > | Qwen3-32B          | 87.8          | 39.2 (-5)            | 31.6 (-7)             | NA                | NA                 |
> > | Qwen3-30B          | 89.0          | 48.6 (-3)            | 42.8 (-1)             | NA                | NA                 |
> > | R1-Qwen-32B        | 89.7          | 37.4 (-8)            | 39.0 (-5)             | NA                | NA                 |
> > | AM-Thinking-32B    | 89.7          | 36.0 (-10)           | 33.7 (-9)             | NA                | NA                 |
> > | QwQ-32B            | 90.5          | 42.3 (-8)            | 38.0 (-8)             | NA                | NA                 |
> >
> > (The +/- change in the parentheses represents in the change in rank compared to benchmark avg.)

---

> ### Author Response · Authors · 2025-11-24
> **Official Response to Review BJqx Part 2**
>
> **[Response to Question 1]**
>
> Our paper consists of two parts. In the first part, we focus on off-the-shelf evaluation of SOTA open-weight reasoning LLMs.
>
> Our experiments in the second part of the paper (Section 4: Control Study) already addresses the concern you mention. To ensure apples-to-apples comparisons between models, we train our own models on fully open-source datasets. Particularly, we use the questions from the open dataset MATH-8K (Zeng, Weihao, et al., 2025) and distill trajectories from three teacher models (QwQ-32B, AM-Thinking-32B, and Qwen3-32B). We will make all these datasets public in the non-anonymized version of our paper.
>
> This allows us to conduct controlled experiments and isolate effects of SFT vs RL, teacher model used for distillation, and data quality filtering heuristics. Note that we cannot train large models, e.g. in the scale of 32B as with the strongest open-source models due to compute restrictions, so we conduct our thorough analysis to smaller models.
>
> **[Response to Question 2]**
> We would like to clarify two things.
>
> First, many models in our evaluation experiments (Table 1) are, in fact, trained with RL (e.g., QwQ-32B, Qwen3-32B, AM-Thinking-32B, DeepScaleR-1.5B, and DeepMath-1.5B).
>
> Second, we dedicate the entire Section 4.2 to study the impact of RL on Recoverability and Guidability in control study. Figure 6 shows the main results from our controlled RL training. We observe that RL substantially improves model’s recoverability and guidability. Section 4.2 provides a detailed breakdown of these effects.
>
> **We hope that our clarifications and follow-up experiment results address the concerns. If the reviewer find these satisfactory, we would be grateful if you would consider raising their score to reflect the updates. We are happy to answer any additional questions and provide clarifications.**
>
> References
>
> - Yang, Wang, et al. "Speculative thinking: Enhancing small-model reasoning with large model guidance at inference time." arXiv preprint arXiv:2504.12329 (2025).
>
> -  Akhauri, Yash, et al. "Splitreason: Learning to offload reasoning." arXiv preprint arXiv:2504.16379 (2025).
>
> - Sohee Yang, Sang-Woo Lee, Nora Kassner, Daniela Gottesman, Sebastian Riedel, and Mor Geva. 2025. How Well Can Reasoning Models Identify and Recover from Unhelpful Thoughts?. In Findings of the Association for Computational Linguistics: EMNLP 2025, pages 7030–7047, Suzhou, China. Association for Computational Linguistics.
>
> -  Zhou, Zhanke, et al. "Can language models perform robust reasoning in chain-of-thought prompting with noisy rationales?." Advances in Neural Information Processing Systems 37 (2024): 123846-123910.
>
> - Zeng, Weihao, et al. "Simplerl-zoo: Investigating and taming zero reinforcement learning for open base models in the wild." arXiv preprint arXiv:2503.18892 (2025).
>
> - Liu, Zichen, et al. "Understanding r1-zero-like training: A critical perspective." arXiv preprint arXiv:2503.20783 (2025).
>
> - Yu, Qiying, et al. "Dapo: An open-source llm reinforcement learning system at scale." arXiv preprint arXiv:2503.14476 (2025).
>
> - Liu, Zihan, et al. "Acemath: Advancing frontier math reasoning with post-training and reward modeling." Findings of the Association for Computational Linguistics: ACL 2025. 2025.
>
> - Zelikman, Eric, et al. "Star: Bootstrapping reasoning with reasoning." Advances in Neural Information Processing Systems 35 (2022): 15476-15488.
>
> - Zelikman, Eric, et al. "Quiet-STaR: Language Models Can Teach Themselves to Think Before Speaking." CoRR (2024).

---

### Official Review · Reviewer_zPNH · 2025-10-31

**Soundness:** 3
**Presentation:** 2
**Contribution:** 2
**Rating:** 4
**Confidence:** 3

**Summary:**

This paper systematically investigates the capabilities of large language models in multi-model collaborative reasoning, introducing the novel paradigm of "off-trajectory reasoning." The authors design a dual-test framework of Recoverability and Guidability, revealing that current models trained for solo reasoning exhibit fragility in collaborative scenarios - stronger benchmark performers are more susceptible to interference, and all models struggle to effectively leverage external correct reasoning traces. This work exposes limitations in conventional training paradigms and provides both evaluation tools and improvement directions for building genuinely collaborative reasoning systems.

**Strengths:**

1. Innovative methods: The paper's innovative dual-test framework of Recoverability and Guidability provides the first systematic tools for evaluating LLMs in collaborative reasoning. This novel and practical methodology effectively uncovers model deficiencies that remain hidden in traditional solo-reasoning benchmarks.
2. Rigorous experiments: Through extensive experiments across multiple models and tasks, the paper presents a counterintuitive finding: strong solo-reasoning models do not equate to strong collaborators. This reveals an orthogonal relationship between benchmark performance and collaborative robustness, posing a significant challenge to the prevailing training paradigm that over-optimizes for leaderboard rankings.
3. Controlled variable experiments: Through controlled variable experiments, the study clearly demonstrates how teacher model selection, RL training, and data filtering strategies directly impact collaborative reasoning capabilities. These findings provide concrete guidance for industrial practitioners to optimize model training.

**Weaknesses:**

1. Limitations of the experimental scope: The paper's experiments are confined to mathematical reasoning tasks, limiting the generalizability of its findings to other collaborative domains like coding or scientific inquiry. Furthermore, the evaluation relies on simulated two-model interactions rather than testing in real multi-agent environments, leaving its practical efficacy unverified.
2. The analysis of model defects is relatively shallow: While the study successfully identifies failures in collaborative reasoning, it does not delve into the underlying causes. It remains unclear whether models fail to "recognize the relevance of guidance" or "cannot integrate it into their own reasoning process," and the analysis lacks mechanistic investigations (e.g., attention weights or intermediate step tracing) to pinpoint these cognitive bottlenecks.

**Questions:**

1. When constructing distracting reasoning traces, did the authors consider incorporating statements with logical fallacies as distractors, in addition to sampling from different questions? This would help evaluate the models' ability to discern and reject fundamentally unsound reasoning.
2. There is a spelling error in the comments below Table 1 on page 5. The word "bechmark" in "the best bechmark model" should be corrected to "benchmark".

---

> ### Author Response · Authors · 2025-11-22
> **Official Response to Reviewer zPNH Part 1**
>
> **[Response to Weakness 1]**
>
> In our work, we deliberately focused on math benchmarks for conducting twin tests as the majority of recent reasoning work and artifacts in open-source community (DeepMath, DeepScaleR, LIMO-32B) have exclusively trained on math data. This focus is advantageous because (a) there exist more open-weight model artifacts in math domain for us to study off-trajectory reasoning (e.g., LIMO-32B frequently fails to generate CoT thinking for non-math questions); (b) many prior works in LLM reasoning (Liu Zichen et al., 2025; Dapo et al., 2025; Liu Zihan et al., 2025; Zelikman, 2022; Zelikman, 2024 ) develop and validate their proposed methods or training recipes primarily on math tasks.  In addition, prior work (e.g., Sprague et al.) shows that scientific-inquiry benchmarks (e.g., GPQA, MMLU) benefit much less from long CoT, as these tasks are predominantly knowledge-based, and even when reasoning is involved, it is often mathematical in nature.
>
> Nevertheless, we did conduct additional experiments in coding. We run our Recoverability and Guidability twin tests on four benchmarks (MBPP, HumanEval, CruxEval, EvalPlus). ***The results are reported in the table below.** We find that the findings from our paper in the math domain hold on the code domain as well: (1) models with stronger solo reasoning do not consistently achieve higher recoverability, and (2) guidability is not strictly correlated with benchmark strength. Interesting, the AM-Thinking-32B, which we specially highlighted as a model with a strong benchmark scores and very poor recoverability on math reports a similar gap between on the code domain as well. This highlights an inherent weakness of this model that is not reflected in standard benchmarks but is exposed via our twin tests.
>
> Since our central findings transfer to coding tasks, we believe this addresses the generalization concern. As for creative writing task, not only is it difficult to reliably evaluate and verify without human efforts, but also most open-weight models are not trained to reason about writing.
>
> As for the reviewer’s request for testing in real multi-agent environments, our framework is intentionally designed to be simple, reproducible, and controllable. Note that our simple benchmark already helps identify key weaknesses of current LLMs not captured by standard benchmarks. As with any benchmark, we expect that as multi-model collaboration capabilities improve, new tests that capture richer multi-agent interactions will be needed. However, our  twin tests suffice for the current off-the-shelf models.
>
>
> | Model              | Benchmark Avg. | Recoverability (Sh.) | Recoverability (Ind.) | Guidability (Sh.) | Guidability (Ind.) |
> |-------------------|----------------|-----------------------|------------------------|--------------------|---------------------|
> | R1-Qwen-1.5B       | 53.3          | 23.0 (+0)            | 17.9 (+0)             | 28.0 (+2)         | 52.8 (+2)          |
> | OpenThinker-1.5B   | 53.4          | 59.2 (+10)           | 47.6 (+10)            | 37.3 (+2)         | 53.4 (+2)          |
> | DeepScaleR-1.5B    | 58.8          | 45.9 (+4)            | 34.3 (+2)             | 27.1 (-2)         | 42.7 (-1)          |
> | DeepMath-1.5B      | 59.8          | 53.8 (+6)            | 40.6 (+4)             | 27.4 (-2)         | 42.2 (-3)          |
> | Qwen3-1.7B         | 74.7          | 70.2 (+9)            | 52.1 (+8)             | 44.4 (+0)         | 56.6 (+1)          |
> | R1-Llama-8B        | 76.0          | 50.1 (+3)            | 43.7 (+5)             | 48.2 (+1)         | 57.1 (+1)          |
> | R1-Qwen-7B         | 77.7          | 35.2 (-5)            | 27.5 (-5)             | 47.9 (-1)         | 55.7 (-2)          |
> | OpenThinker-7B     | 80.1          | 68.4 (+5)            | 61.6 (+6)             | NA                | NA                 |
> | Qwen3-8B           | 87.4          | 54.4 (+2)            | 42.2 (+0)             | NA                | NA                 |
> | Qwen3-32B          | 87.8          | 39.2 (-5)            | 31.6 (-7)             | NA                | NA                 |
> | Qwen3-30B          | 89.0          | 48.6 (-3)            | 42.8 (-1)             | NA                | NA                 |
> | R1-Qwen-32B        | 89.7          | 37.4 (-8)            | 39.0 (-5)             | NA                | NA                 |
> | AM-Thinking-32B    | 89.7          | 36.0 (-10)           | 33.7 (-9)             | NA                | NA                 |
> | QwQ-32B            | 90.5          | 42.3 (-8)            | 38.0 (-8)             | NA                | NA                 |
>
> (The +/- change in the parentheses represents in the change in rank compared to benchmark avg.)

---

> ### Author Response · Authors · 2025-11-22
> **Official Response to Reviewer zPNH Part 2**
>
> **[Response to Weakness 2]**
> In addition to our analysis of reasoning data and detailed ablation study in the original paper, we include additional mechanistic analysis below.
>
> 1. Mechanistically understanding the recoverability results: attention-based recovery signals
>
> We conduct follow-up analysis to better understand why models fail or succeed at recovering from distracting steers. As suggested, we study attention patterns to answer this question. Our hypothesis is that successful recovery happens when attention heads allocate *more weight to the original problem tokens* during continued reasoning, rather than the *distracting steer tokens*.
>
> To test this, we extract full attention matrices (layer × head) over sequences of the form **[problem → distracting steer → continued reasoning]** from the recoverability experiment for each model. Then, for reasoning tokens in the continued reasoning, we aggregate the attention mass assigned to (1) problem, (2) distractor, and (3) continued reasoning tokens. This yields a *(num_layers × num_heads, 3) representation* for each successful and unsuccessful recovery example.
>
> We investigate if these representations alone, capturing only accumulated attention mass over three different regions, can predict the success or failure in the recoverability test. We train Lasso classifiers with 5-fold cross-validation for each model separately. We find that, across reasoning LLMs, the regression models achieve 70–80% balanced test accuracy at predicting recoverability success. Digging deeper, we found that ~5 attention heads per model could be consistently identified whose attention distribution is strongly predictive of recovery success. Importantly, by computing Cohen’s d (i.e., standardized mean difference  between the attention distributions of successful and unsuccessful recoveries), we find that higher attention to the problem region vs. distractor region is associated with successful recovery—supporting our hypothesis that models allocate more attention to the problem region for successful recoveries.
>
> These are some examples of Lasso regression results. It shows that the attention features can predict distraction recovery success with high accuracy. We will include more detailed results in the updated paper.
>
> | Split               | R1-Distill-Qwen-7B | R1-Distill-Llama-8B | QwQ-32B        | Qwen3-30B      |
> |---------------------|--------------------|----------------------|----------------|----------------|
> | Cross-Validation    | 71.0 ± 5.6         | 81.0 ± 4.6           | 81.5 ± 4.4     | 78.0 ± 7.8     |
> | Baseline (Majority) | 51.7               | 65.0                 | 53.3           | 68.3           |
>
> For R1-Distill-Qwen-7B, as an example, we found two stable attention heads, Layer23Head17 and L24H0, where attention to the distractor by the former indicates unsuccessful recovery, and attention to the problem by the latter indicates success. We will include full results to the updated paper.

---

> > ### Author Response · Authors · 2025-11-22
> > **Official Response to Reviewer zPNH Part 2 Continued**
> >
> > 2. Whether models fail to "recognize the relevance of guidance" or "cannot integrate it into their own reasoning process?"
> >
> > We are not confident we understand this comment. Particularly, it is unclear how to robustly measure whether LLMs “recognize the relevance of guidance” or “integrate” it.
> >
> > We can, however, provide anecdotal examples from our qualitative analysis. Let us loosely define that LLMs “recognize the relevance of guidance” if they coherently continue the reasoning of the guiding steer. Note that the continuation can be coherent even when it is incorrect. **Overall, we observe that models do coherently continue the reasoning in the provided guiding steer for both successful and unsuccessful cases; they do not “ignore” the guide, immediately backtrack or degenerate in obvious ways.** So, in this sense, models do “recognize the relevance of the guidance”. However, these can still lead to incorrect solutions, showcasing a failure to “integrate” or build upon these correct guiding steers.
> >
> > For instance, this is a correct continued reasoning from guidability:
> > ```
> > **Problem:** Determine the energy gap (in eV) between the electronic states $n=7$ and $n=8$ in hydrogen. Please format your answer as $n \times 10^x$ where $n$ is to 1 decimal place.
> >
> > **Guidance**: […] But let me check if there's a more precise way. Let me compute 1/49 - 1/64 exactly.The difference is (64 - 49)/(49*64) = 15/(3136) = 15/3136 ≈ 0.004783163\nMultiply by 13.6:\n15/3136 * 13.6 = (15*13.6)/3136\n15*13.6 = 204\n204/3136 ≈ 0.06505102 eV
> >
> > **Continued Reasoning**: Wait, but in this case, shifting in x from (0,1) gives (1,1)=4, and shifting in x from (1,0) gives (2,0)=3, so 4>3, which is consistent with 2>1. [...]
> > ```
> > This is an incorrect case:
> > ```
> >
> > **Problem:** Find the theoretical limiting angular resolution (in arcsec) of a commercial 8-inch (diameter) optical telescope being used in the visible spectrum (at $\lambda=5000 \AA=500 \mathrm{~nm}=5 \times 10^{-5} \mathrm{~cm}=5 \times 10^{-7} \mathrm{~m}$).  Answer in arcseconds to two significant figures.
> >
> > **Guidance**: […] I can use consistent units. Sometimes people use λ and D in the same units.Let me try in centimeters. λ = 500 nm = 5e-5 cm? 1 nm = 10^{-9} m, 1 cm = 0.01 m, so 1 nm = 10^{-7} cm? Let's see. 1 m = 100 cm, so 1 nm = 10^{-9} m = 10^{-9} * 100 cm = 10^{-7} cm? 10^{-9} * 100 = 10^{-7}, yes.
> >
> > **Continued Reasoning**: So 500 nm = 500e-7 cm = 5e-5 cm. D = 8 inches. 1 inch = 2.54 cm, so 8 inches = 8 * 2.54 cm = 20.32 cm. So λ/D = 5e-5 cm / 20.32 cm = 2.46e-6 radians, same as before. [...]
> > ```
> >
> > Additionally, we test whether successful guidance is more **in-distribution** for the main model compared to unsuccessful ones. Here, we assume that an in-distribution guide may lead to better “recognition”. We conduct a likelihood analysis: for each instance, we compute the per-token average probability (or inverse perplexity) of the *guidance region* conditioned on the problem.If successful guidances are better “recognized,” they should yield higher likelihood.
> >
> > Across models, we do **not** observe a consistent difference. Only four models (DeepScaleR-1.5B, OpenThinker3-1.5B, R1-Llama-8B, LIMO-32B) show significantly higher likelihood (p < 0.05) for successful guiding steers compared to unsuccessful. We posit it is due to model’s inherent inability to understand and continue from the guidance to find the correct answer.
> >
> > | Model               | Correct Likelihood | Incorrect Likelihood | P-value   |
> > |---------------------|--------------------|-----------------------|-----------|
> > | R1-Qwen-1.5B        | 0.44               | 0.42                  | 0.11      |
> > | DeepScaleR-1.5B     | 0.41               | 0.39                  | 0.02      |
> > | R1-Llama-8B         | 0.47               | 0.43                  | 1.2e-05   |
> > | DeepMath-1.5B       | 0.40               | 0.39                  | 0.07      |
> > | OpenThinker3-1.5B   | 0.40               | 0.39                  | 0.45      |
> > | Qwen3-1.7B          | 0.34               | 0.32                  | 0.04      |
> > | R1-Qwen-7B          | 0.44               | 0.35                  | 0.10      |
> > | LIMO-32B            | 0.47               | 0.45                  | 0.0008    |
> > | R1-Qwen-32B         | 0.50               | 0.49                  | 0.195     |

---

> ### Author Response · Authors · 2025-11-22
> **Official Response to Reviewer zPNH Part 3**
>
> **[Response to Question 1: test on fallacies as distractors]**
>
> This is a good point! We did, initially, consider designing steers that introduce logical fallacies. However, different LLMs have distinct reasoning behaviors and different benchmark questions require different kinds of reasoning. Therefore, it is extremely challenging to devise a protocol to inject logical fallacies in a scalable manner.
>
> Prior works (Lanham et al., 2023; ) have often used simple corruptions to introduce trivial logical fallacies (e.g., “1+2=3” → “1+2=4”, or digit randomization) to test robustness of previous generation of LLMs. However, we found that such trivial digit corruptions do not serve as an effective test for current models, particularly reasoning models, which are largely robust to such local corruptions.
>
> To convincingly demonstrate this, we implement a controlled *calculation fallacy* setup: we identify numbers in the ongoing reasoning and inject corruptions (e.g., “12” → “112”, “12” → “21”, or digit randomization).
>
> We find that most models have a >80% recovery rate when less than 15 digits are corrupted. Moreover, these recovery rates are highly correlated with benchmark performance—unsurprising, since robust math models routinely backtrack and verify intermediate steps. In contrast, the distraction-recovery behavior studied in our paper is *not* explained by benchmark scores and captures a distinct, under-examined failure mode.
>
> | Model                   | Digit Corruption Recovery | Benchmark Avg. |
> |-------------------------|------------------|----------------|
> | R1-Distill-Qwen-1.5B    | 63.1             | 47.5           |
> | R1-Distill-Qwen-7B      | 69.7             | 64.6           |
> | R1-Distill-Llama-8B     | 97.3             | 54.1           |
> | QwQ-32B                 | 98.3             | 80.5           |
> | R1-Distill-Qwen-32B     | 78.8             | 72.3           |
> | AM-Distill-Qwen-32B     | 87.8             | 82.6           |
> | Qwen3-1.7B              | 96.8             | 59.9           |
> | Qwen3-8B                | 97.3             | 79.1           |
> | Qwen3-30B-A3B           | 96.6             | 81.1           |
> | Qwen3-32B               | 94.5             | 81.0           |
>
>
> **[Response to Question 2]**
>
> Thank you! We will fix all the typos in the paper.
>
> **If the reviewer finds these satisfactory, we respectfully ask that they consider raising their score to reflect these clarifications and improvements.**
>
> **References**
> - Sprague, Zayne Rea, et al. "To CoT or not to CoT? Chain-of-thought helps mainly on math and symbolic reasoning." The Thirteenth International Conference on Learning Representations.
>
> - Lanham, Tamera, et al. "Measuring Faithfulness in Chain-of-Thought Reasoning." CoRR (2023).

---

### Official Review · Reviewer_BaPD · 2025-11-03

**Soundness:** 3
**Presentation:** 3
**Contribution:** 3
**Rating:** 6
**Confidence:** 2

**Summary:**

This paper investigates whether reasoning LLMs trained for solo reasoning can effectively collaborate by working on shared reasoning trajectories—a capability the authors term "off-trajectory reasoning." The authors propose twin evaluation tests: (1) **Recoverability**, which measures whether LLMs can backtrack from distracting reasoning traces, and (2) **Guidability**, which tests whether they can build upon correct partial reasoning from stronger models to solve problems beyond their inherent capabilities. The study evaluates 15 open-weight LLMs (1.5B–32B parameters) across five math benchmarks (AIME2024/2025, MATH-500, Minerva, OlympiadBench).

**Strengths:**

- First systematic study examining whether solo-reasoning training yields collaboration-ready models
- Comprehensive evaluation across 15 open-weight LLMs from 4 distinct families
- Clear problem motivation connecting to practical collaboration scenarios (efficiency, exploration, safety)
- Provides actionable insights for training better collaborative reasoners

**Weaknesses:**

- All experiments confined to mathematical reasoning benchmarks
- The assumption that reasoning for question q' inserted into question q constitutes a "strong distractor" is not validated
- No analysis of what specific data characteristics lead to poor/variable recoverability
- No discussion of computational costs of multi-model collaboration vs. solo reasoning

**Questions:**

- How do results change if steers are from different model families rather than the same model?
- Can you show cases where models successfully integrate guidance? What makes them different?
- Does synthetic data generation for off-trajectory scenarios help?

---

> ### Author Response · Authors · 2025-11-24
> **Official Response to Reviewer BaPD Part 1**
>
> **[Response to Weakness 1]**
>
> In our work, we deliberately focused on math benchmarks for conducting twin tests as the majority of recent reasoning work and artifacts in open-source community (DeepMath, DeepScaleR, LIMO-32B) have exclusively trained on math data. This focus is advantageous because (a) there exist more open-weight model artifacts in math domain for us to study off-trajectory reasoning (e.g., LIMO-32B frequently fails to generate CoT thinking for non-math questions); (b) many prior works in LLM reasoning (Liu Zichen et al., 2025; Dapo et al., 2025; Liu Zihan et al., 2025; Zelikman, 2022; Zelikman, 2024 ) develop and validate their proposed methods or training recipes primarily on math tasks.  In addition, prior work (e.g., Sprague et al.) shows that scientific-inquiry benchmarks (e.g., GPQA, MMLU) benefit much less from long CoT, as these tasks are predominantly knowledge-based, and even when reasoning is involved, it is often mathematical in nature.
>
> Nevertheless, we did conduct additional experiments in coding. We run our Recoverability and Guidability twin tests on four benchmarks (MBPP, HumanEval, CruxEval, EvalPlus). The results are reported in the table below. We find that the findings from our paper in the math domain hold on the code domain as well: (1) models with stronger solo reasoning do not consistently achieve higher recoverability, and (2) guidability is not strictly correlated with benchmark strength. Interesting, the AM-Thinking-32B, which we specially highlighted as a model with a strong benchmark scores and very poor recoverability on math reports a similar gap between on the code domain as well. This highlights an inherent weakness of this model that is not reflected in standard benchmarks but is exposed via our twin tests. Since our central findings transfer to coding tasks, we believe this addresses the generalization concern.
>
> | Model              | Benchmark Avg. | Recoverability (Sh.) | Recoverability (Ind.) | Guidability (Sh.) | Guidability (Ind.) |
> |-------------------|----------------|-----------------------|------------------------|--------------------|---------------------|
> | R1-Qwen-1.5B       | 53.3          | 23.0 (+0)            | 17.9 (+0)             | 28.0 (+2)         | 52.8 (+2)          |
> | OpenThinker-1.5B   | 53.4          | 59.2 (+10)           | 47.6 (+10)            | 37.3 (+2)         | 53.4 (+2)          |
> | DeepScaleR-1.5B    | 58.8          | 45.9 (+4)            | 34.3 (+2)             | 27.1 (-2)         | 42.7 (-1)          |
> | DeepMath-1.5B      | 59.8          | 53.8 (+6)            | 40.6 (+4)             | 27.4 (-2)         | 42.2 (-3)          |
> | Qwen3-1.7B         | 74.7          | 70.2 (+9)            | 52.1 (+8)             | 44.4 (+0)         | 56.6 (+1)          |
> | R1-Llama-8B        | 76.0          | 50.1 (+3)            | 43.7 (+5)             | 48.2 (+1)         | 57.1 (+1)          |
> | R1-Qwen-7B         | 77.7          | 35.2 (-5)            | 27.5 (-5)             | 47.9 (-1)         | 55.7 (-2)          |
> | OpenThinker-7B     | 80.1          | 68.4 (+5)            | 61.6 (+6)             | NA                | NA                 |
> | Qwen3-8B           | 87.4          | 54.4 (+2)            | 42.2 (+0)             | NA                | NA                 |
> | Qwen3-32B          | 87.8          | 39.2 (-5)            | 31.6 (-7)             | NA                | NA                 |
> | Qwen3-30B          | 89.0          | 48.6 (-3)            | 42.8 (-1)             | NA                | NA                 |
> | R1-Qwen-32B        | 89.7          | 37.4 (-8)            | 39.0 (-5)             | NA                | NA                 |
> | AM-Thinking-32B    | 89.7          | 36.0 (-10)           | 33.7 (-9)             | NA                | NA                 |
> | QwQ-32B            | 90.5          | 42.3 (-8)            | 38.0 (-8)             | NA                | NA                 |
>
> (The +/- change in the parentheses represents in the change in rank compared to benchmark avg.)

---

> ### Author Response · Authors · 2025-11-24
> **Official Response to Reviewer BaPD Part 2**
>
> **[Response to Weakness 2]**
>
> Empirically, most of the off-the-shelf models in our results in Table 1 shows low recoverability. **This, by itself, provides some evidence that our chosen distractors are “strong” enough for the current generation of models we target in our study.** We do not claim that stronger distractors cannot be obtained, only that ours are strong enough for us to demonstrate the mismatch between benchmark scores and recoverability capabilities for current models.
>
> Note that by sampling from a different question $q’$, we can guarantee that the distractor reasoning is incorrect. Alternatively, if we defined our distractor steers to be a partial trajectory from one sampled for $q$ but is incorrect, we could not have guaranteed that the partial trajectory contains an error and is in fact a distractor without further processing, say by using an LLM judge. Therefore, sampling from $q’$ also has the added benefit of scalability and cost efficiency.
>
> Let us also expand on our reasons for our distractor sampling strategy. It originated from the observation that incomplete reasoning traces can often mislead LLMs to continue the trace. This is similar to how LLMs sometimes follow user instructions even when they contradict the system instructions earlier in its prefix, despite their safety training (Anil, Cem et al., 2024; Zou, Andy et al., 2023). Our distractor strategy also shares similarities with recently published work (Yang et al., 2025) that studies meta-cognitive behaviors of reasoning models and uses a very similar method for constructing “uninformative”, “irrelevant”, or “unhelpful” thoughts to perturb model trajectories.
>
> References:
> - Anil, Cem, et al. "Many-shot jailbreaking." Advances in Neural Information Processing Systems 37 (2024): 129696-129742.
> - Zou, Andy, et al. "Universal and transferable adversarial attacks on aligned language models." arXiv preprint arXiv:2307.15043 (2023).
>
> **[Response to Weakness 3]**
>
> We discuss this briefly in our main results. Particularly, Finding 2 identifies the re-statement of the problem at the beginning of the thinking to be critical to problem solving. Recall that most reasoning LLMs tend restate the problem in the beginning of its thinking process before actual problem solving. Our analysis has revealed that LLMs are most vulnerable to distraction at the beginning of thinking, likely because LLMs anchor their later reasoning on this problem restatement instead of the problem in the user message.
>
> Additionally, we investigate whether the recoverability rate of a model on a particular question is correlated with the model’s solve rate (SR) for that question. The table below reports the recoverability for a subset of models at different solve rates (the full table is in Appendix D, Table 9). As the table shows, we observe that model’s distractibility is highest for questions with a lower SR. This is despite the fact that we always choose correct trajectory to insert a distractor in and conduct our analysis. We will add these additional analysis to the paper.
>
> | Model            | SR=1/8 | SR=2/8 | SR=3/8 | SR=4/8 | SR=5/8 | SR=6/8 | SR=7/8 | SR=8/8 | Avg. |
> |------------------|--------|--------|--------|--------|--------|--------|--------|--------|------|
> | R1-Qwen-1.5B     | 16.1   | 27.7   | 28.6   | 33.8   | 36.5   | 40.0   | 48.1   | 58.9   | 38.6 |
> | Qwen3-1.7B       | 40.9   | 38.0   | 52.3   | 65.2   | 70.6   | 80.6   | 84.7   | 91.3   | 74.6 |
> | R1-Llama-8B      | 30.5   | 23.3   | 39.4   | 51.8   | 56.4   | 47.9   | 65.2   | 70.0   | 49.6 |
> | R1-Qwen-32B      | 24.7   | 35.4   | 40.0   | 40.0   | 32.5   | 41.9   | 48.0   | 52.9   | 45.6 |
> | AM-Thinking-32B  | 11.1   | 22.2   | 27.5   | 12.5   | 23.6   | 16.7   | 21.4   | 28.9   | 25.3 |

---

> ### Author Response · Authors · 2025-11-24
> **Official Response to Reviewer BaPD Part 3**
>
> **[Response to Weakness 4]**
>
> That’s a good question. In this paper, our focus is not to improve the accuracy-efficiency trade-offs with multi-model collaboration compared to solo reasoning. Recent works already establish that this trade-off can be improved with multi-model collaboration, e.g. SplitReason (Akhauri et al., 2025) and Speculative Thinking (Yang, Wang, et al., 2025), which train two-model collaboration systems. As an example, Yang, Wang, et al. show that their collaborative system (between a 1.5B and a 32B model) leads to 170% to 200% speedup on math benchmarks with 5.0% to 8.1% improvement on performance.
>
> We agree with these papers and believe that multi-model collaboration, especially using the mechanism we describe, offers efficiency benefits amongst others. But our focus is different: we want to study the challenges that emerge due reasoning conditioned on **out-of-distribution trajectories** in a multi-model collaborative setting. Moreover, the costs of multi-model collaboration depend on the relative sizes of the two models, and the collaboration mechanism. In our work, we are able to re-use guide steers from the stronger models to analyze many different weaker models, and therefore, our computational costs do not reflect an online collaborative setup.
>
>
> **[Response to Question 1]**
>
> Thanks for bringing up this question. For guidability, we have reported the aggregate performance in Table 1 using guiding steers source from 3 different models (DeepSeek-R1, QwQ-32B and Qwen3-235B-A22B). The guidability performance per source is reported in Table 12 of  Appendix E. We find that models, on average, report the best guidability results when conditioning on guiding steers from the QwQ-32B model.
>
> Furthermore, as requested by the reviewer, we conduct a similar ablation study on recoverability by sampling distractors from 5 different models, and compare these results to the original recoverability results reported in Table 1 of the paper.  As shown in the table below, the recoverability performance is similar for the two cases — when the distractor is sampled from the same versus an external model. This further shows that our protocol design yields reliable evaluation for model’s robustness to recover from distraction.
>
> We also want to highlight that, if the collaborator model(s) are known a priori, our test protocol can be easily generalized to sampling from these models to test distractibility. However, if they are unknown, our protocol provides a model-agnostic and simple way to reliably measure an LLM’s recoverability.
>
> | Model               | Recoverability (same model) | Recoverability (different models) |
> |---------------------|-----------------------------|-----------------------------------|
> | R1-Qwen-1.5B        | 60.6                        | 68.8                              |
> | DeepScaleR-1.5B     | 82.4                        | 80.7                              |
> | R1-Llama-8B         | 81.4                        | 79.9                              |
> | DeepMath-1.5B       | 88.0                        | 92.7                              |
> | OpenThinker3-1.5B   | 95.2                        | 91.7                              |
> | Qwen3-1.7B          | 98.4                        | 97.5                              |
> | R1-Qwen-7B          | 73.5                        | 70.0                              |
> | LIMO-32B            | 29.3                        | 25.2                              |
> | OpenThinker3-7B     | 85.6                        | 85.5                              |
> | R1-Qwen-32B         | 69.8                        | 66.1                              |
> | Qwen3-8B            | 85.9                        | 85.6                              |
> | QwQ-32B             | 79.9                        | 77.8                              |
> | Qwen3-32B           | 71.8                        | 73.8                              |
> | Qwen3-30B-A3B       | 87.8                        | 86.7                              |
> | AM-Thinking-32B     | 33.4                        | 33.9                              |

---

> ### Author Response · Authors · 2025-11-24
> **Official Response to Reviewer BaPD Part4**
>
> **[Response to Question 2]**
>
> That’s a good question. In our qualitative analysis, we observed that model’s continuations are generally coherent and attempt to build upon the teachers reasoning for both successful and unsuccessful scenarios. They do not “ignore” the guide, immediately backtrack or degenerate in obvious ways.  Here, we provide an example to compare successful vs unsuccessful cases that can represent our overall qualitative analysis. Given the overall low guidability scores for math, we hypothesize that the models are largely incapable of following and leveraging off-trajectory guides.
>
> For instance, this is a correct continued reasoning from guidability:
> ```
> **Problem:** Determine the energy gap (in eV) between the electronic states $n=7$ and $n=8$ in hydrogen. Please format your answer as $n \times 10^x$ where $n$ is to 1 decimal place.
>
> **Guidance**: […] But let me check if there's a more precise way. Let me compute 1/49 - 1/64 exactly.The difference is (64 - 49)/(49*64) = 15/(3136) = 15/3136 ≈ 0.004783163\nMultiply by 13.6:\n15/3136 * 13.6 = (15*13.6)/3136\n15*13.6 = 204\n204/3136 ≈ 0.06505102 eV
>
> **Continued Reasoning**: Wait, but in this case, shifting in x from (0,1) gives (1,1)=4, and shifting in x from (1,0) gives (2,0)=3, so 4>3, which is consistent with 2>1.
> ```
> This is an incorrect case:
> ```
>
> **Problem:** Find the theoretical limiting angular resolution (in arcsec) of a commercial 8-inch (diameter) optical telescope being used in the visible spectrum (at $\lambda=5000 \AA=500 \mathrm{~nm}=5 \times 10^{-5} \mathrm{~cm}=5 \times 10^{-7} \mathrm{~m}$).  Answer in arcseconds to two significant figures.
>
> **Guidance**: […] I can use consistent units. Sometimes people use λ and D in the same units.Let me try in centimeters. λ = 500 nm = 5e-5 cm? 1 nm = 10^{-9} m, 1 cm = 0.01 m, so 1 nm = 10^{-7} cm? Let's see. 1 m = 100 cm, so 1 nm = 10^{-9} m = 10^{-9} * 100 cm = 10^{-7} cm? 10^{-9} * 100 = 10^{-7}, yes.
>
> **Continued Reasoning**: So 500 nm = 500e-7 cm = 5e-5 cm. D = 8 inches. 1 inch = 2.54 cm, so 8 inches = 8 * 2.54 cm = 20.32 cm. So λ/D = 5e-5 cm / 20.32 cm = 2.46e-6 radians, same as before.
> ```
>
> **[Response to Question 3]**
>
> Thanks for bringing up this question. We deliberately avoid any post-training that directly targets our twin tests. If we generate synthetic data, such as successful recovery or RL on off-trajectory scenarios, to optimize model’s reasoning for the twin tests, we would expect to see improvements on these metrics. However, we think this defeats the original purpose of our tests, which aim to evaluate off-trajectory reasoning of off-the-shelf models. As the Goodhart's Law says, “when a measure becomes a target, it ceases to be a good measure.” This means that setting a specific metric as a goal creates an incentive to manipulate that metric, often at the expense of the original, broader objective.
>
> In our work, the broader goal of our recoverability is to measure model’s robustness in its reasoning when perturbed off their training distribution. The distraction in real-world test time settings can be in different forms (all of which cannot be enumerated or incorporated in training), and our protocol is a simple yet effective way to simulate the general “distracting effects” of any model reasoning intervention. Yet, this does not mean we should directly optimize against this metric, as it may misalign with the overall goal and overfit to our specific distractors.
>
> **We hope that our explanations and follow-up experiment results address the concerns raised by the reviewer. If the reviewer finds these satisfactory, we respectfully ask that they consider raising their score to reflect these updates.**
>
> References:
> - Yang, Wang, et al. "Speculative thinking: Enhancing small-model reasoning with large model guidance at inference time." arXiv preprint arXiv:2504.12329 (2025).
>
> - Akhauri, Yash, et al. "Splitreason: Learning to offload reasoning." arXiv preprint arXiv:2504.16379 (2025).
>
> - Sohee Yang, Sang-Woo Lee, Nora Kassner, Daniela Gottesman, Sebastian Riedel, and Mor Geva. 2025. How Well Can Reasoning Models Identify and Recover from Unhelpful Thoughts?. In Findings of the Association for Computational Linguistics: EMNLP 2025, pages 7030–7047, Suzhou, China. Association for Computational Linguistics.
>
> - Anil, Cem, et al. "Many-shot jailbreaking." Advances in Neural Information Processing Systems 37 (2024): 129696-129742.
>
> - Zou, Andy, et al. "Universal and transferable adversarial attacks on aligned language models." arXiv preprint arXiv:2307.15043 (2023).
>
> - Gao, Leo, John Schulman, and Jacob Hilton. "Scaling laws for reward model overoptimization." International Conference on Machine Learning. PMLR, 2023.
>
> - Dubois, Yann, et al. "Length-controlled alpacaeval: A simple way to debias automatic evaluators." arXiv preprint arXiv:2404.04475 (2024).

---

### Official Review · Reviewer_Y2jS · 2025-11-03

**Soundness:** 3
**Presentation:** 3
**Contribution:** 2
**Rating:** 4
**Confidence:** 3

**Summary:**

The paper proposes two synthetic tests to assess how well LLMs can collaborate on a shared reasoning trajectory, which the authors refer to as off-trajectory reasoning. The tests are as follows:

- Recoverability: tests whether models can backtrack from incorrect reasoning to arrive at the correct answer. This is done by adding a distractor in the middle of a reasoning trace.
  - A distractor is constructed by taking a reasoning path from a *different question* and truncating it to 20% of the target reasoning length.
  - Recoverability tests are performed on questions that the model can already answer correctly.
- Guidability: tests whether the model can successfully use the guidance of a larger guide model to arrive at the correct answer. This is done by adding a portion of the teacher's reasoning at the beginning of the reasoning trace. The length is selected between 20% - 80% the guide's reasoning length.
  - Guidability tests are performed on questions where the model's solve rate is less than 2/8.

The authors perform these tests on a wide variety of models and show that the test metrics are largely orthogonal to the original *solo* reasoning capability of the models. They also perform various ablation on test parameters, and perform control studies on the post-training recipes of the models under evaluation, i.e., SFT distillation and RL.

**Strengths:**

- Novel approach: the authors acutely identify recoverability and guidability as two key pre-requisite abilities for LLM collaboration, and target these aspects through novel probing schemes.
- Simplicity: the two test methodologies are simple and easy to implement
- Novel findings: The main finding is counter-intuitive and novel.
- Actionable insights: The authors present clear, actionable insights from the control studies.

**Weaknesses:**

- W1. Contrived test protocol: the test protocol involves truncating reasoning traces and inserting these partial traces at arbitrary token locations. Also, as I understand, this happens within the internal thinking traces of the models within the, e.g., <think> tags, within the *assistant* message.
  - This introduces misalignment with the training distribution, which is unrelated to the model abilities which the authors intend to evaluate. Firstly, truncating and inserting at arbitrary token indices can lead to incoherent sentences. Also, in standard SFT pipelines, the training data for the *assistant* message portion of the samples are meant to be outputs from the model itself. Furthermore, in RL, the *thinking* part of the assistant message is only trained on outputs generated by the model itself. The proposed test protocol introduces contexts that are significantly misaligned with this training distribution. I.e., placing the external reasoning in the thinking part of the assistant message indicates to the model, perhaps deceivingly, that it originated from the model itself, rather than some external model. The input distribution shift due to the *protocol* will likely cause significant performance degradation, unrelated to the *semantic* distribution shift of distractor reasoning or guidance from an external collaborator LLMs, on which the author *intend* to evaluate the models.
  - While some amount of input distribution shift is inevitable in any inference-time probing scheme, I believe the contrived nature of the  specific approach proposed by the authors may cause excessive distribution shift, which would be avoided in potential real-world methods used for LLM collaboration. For example, it may be possible to include external distractor or guidance reasoning as a user message, indicating the that the reasoning does not originate from the current model. The model will then be inclined to identify potential errors or hints and act accordingly.
- W2. Limited task domain: The study is limited to a single domain, math reasoning. Therefore, the findings may not generalize to other domains such as coding, creative writing, etc., where distractions and guidance may have different characteristics.

**Questions:**

Regarding W1, I'm not sure if we necessarily want or need models to perform well on this test for them to collaborate well with other models. I do not believe the test scores will align with the model's ability when applying realistic inference-time or training-based collaboration methods.

I'm willing to raise my score if the authors can provide a strong argument against this concern.

---

> ### Author Response · Authors · 2025-11-22
> **Response to Reviewer Y2jS Part 1**
>
> **[Response to Weakness 1 & Questions]** Justification for our test protocol: The reviewer raises two primary concerns with the test protocol: (a) The steers in our twin test setup are out-of-distribution, contrasting with the standard reasoning trajectory distribution (both after SFT and RL) which are sampled from the main model itself. The reviewer posits that such steer results in an OOD reasoning prefix and therefore questions the justifications for studying this. (b) Format issues, i.e., our test does not use special tags to distinguish assistant outputs from collaborator outputs, and the trajectories may be incoherent due to injecting distractors at arbitrary positions.
>
> We address both these two points separately below.
>
> **Steers are OOD for the main model**: We argue that this does, in fact, align with how recent work envisions using these reasoning models (Akhauri et al. 2025; Wan, 2025;  Muennighoff et al. 2025; Wu et al., 2025; Yang et al., 2025).
>
> First, as we point out in the paper, multiple works interleave reasoning between two model collaborators, such as weak and strong in SplitReason (Akhauri et al. 2025) and in Speculative Thinking (Yang et al., 2025), a meta-thinking and reasoning models, as in ReMa (Wan, 2025), inject steers like “Wait,” to extend reasoning length beyond the main model’s distribution (e.g. the s1 paper by Muennighoff et al. 2025). In all these prior works, reasoning models continue reasoning from a prefix that is composed of a mix of in- and out-of-distribution sentences.
>
> Moreover, prior work has formalized the idea of directly intervening in the thinking trajectory of the main model as “Thinking Intervention” (Wu et al., 2025), similar to our work. The paper makes a compelling case for using thinking interventions to guide the model’s reasoning in a transparent and controlled manner.
>
> Realistically, as models use tools at inference time to solve long horizon tasks (such as deep research use cases), it is expected that the outputs from these tools will often be out-of-distribution not only for the main model’s output distribution, but also in terms of prefixes it has encountered during training. Therefore, we need models to be robust to inference time out-of-distribution scenarios, the exact capability we study in this paper.
>
> Finally, as the reviewer points out, some amount of distribution shift is inevitable in all inference time probing studies. However, we disagree that ours is “excessive” or that studying this is not well-motivated. In fact, we point to recent published work (Yang et al., 2025) that studies the  robustness of reasoning models by injecting thoughts that are “uninformative”, “irrelevant”, etc., using a direct intervention strategy that is similar to ours in the “degree” of intervention. Importantly, this study does not use special tags to distinguish injected thoughts, similar to our work. We expand more on this format issue below.

---

> ### Author Response · Authors · 2025-11-22
> **Response to Reviewer Y2jS Part 1Continued**
>
> **[Response to Weakness 1 & Questions] Continued**
>
> **Format issues**: We acknowledge the reviewer’s concern that the main model’s and external reasonings can be distinguished using especial tags.
>
> First, in the relevant prior literature, which we discussed above, there exists both works that use special tags to distinguish external reasoning (e.g., SplitReason by Akhauri et al. 2025 and Rema by Wan et al., 2025) and those that directly intervene in the reasoning (Speculative Thinking by Yang et al., 2025; s1 by Muennighoff 2025; Thinking Intervention by Wu et al. 2025, and work by Yang et al. 2025].
>
> Second, in this work, we study off the shelf performance of strong reasoning models released by the open-source community. Therefore, our design choice is also motivated by a practical consideration — we could not introduce new tags (e.g. \<collaborator\> etc.) into the reasoning of off-the-shelf models. This, we argue, would cause degradation due to the *protocol,* as pointed out by the reviewer.
>
> We considered using the \<user\> tags to distinguish between main model and external steer as an alternative, as suggested. However, we run the risk of over-estimating post-trained LLMs’ steerability if the steer is enclosed within user tags as LLMs are especially post-trained to follow user instructions.
>
> We believe that our design choice reflects the best option given these above constraints. We deliberately made these choices to ensure that our steered trajectories are stylistically more similar to standard solo trajectories compared to a case where we inject special tags or use user tags for external reasoning for off-the-shelf models.
>
> The final concern raised is that our strategy results in “incoherent” sentences. In guidability, the steer is always appended at the start of the thinking, and therefore add no incoherence. In recoverability, we keep a coherent prefix (complete sentence) of the original reasoning and then append the distracting reasoning on a new line, prefixed with a short transition such as`"Wait. Let me think"`. Moreover, Figure 4 shows that the recoverability of the models is lowest when the distracting steers are inserted at the 0% position, i.e. at the start of the thinking which does not add any incoherence. This clearly demonstrates that poor recoverability cannot be attributed to incoherent sentences.
>
> We will expand on the description of the twin tests in our paper to explain our design choices more clearly and draw parallels with recent works that use the same protocol as ours for model collaborations. We believe that this will strengthen our paper and thank the reviewer for their questions.

---

> ### Author Response · Authors · 2025-11-22
> **Response to Reviewer Y2jS Part 2**
>
> **[Response to Weakness 2]: Limited task domain**
>
> In our work, we deliberately focused on math benchmarks for conducting twin tests as the majority of recent reasoning work and artifacts in open-source community (DeepMath, DeepScaleR, LIMO-32B) **have exclusively** trained on math data. This focus is advantageous because (a) there exist more open-weight model artifacts in math domain for us to study off-trajectory reasoning (e.g., LIMO-32B frequently fails to generate CoT thinking for non-math questions); (b) many prior works in LLM reasoning (Liu Zichen et al., 2025; Dapo et al., 2025; Liu Zihan et al., 2025; Zelikman, 2022; Zelikman, 2024 ) develop and validate their proposed methods or training recipes primarily on math tasks.  In addition, prior work (e.g., Sprague et al.) shows that knowledge-based benchmarks benefit much less from long CoT, and even when reasoning is involved, it is often mathematical in nature.
>
> Nevertheless, we did conduct additional experiments in coding. We run our Recoverability and Guidability twin tests on four benchmarks (MBPP, HumanEval, CruxEval, EvalPlus). The results are reported in the table below. We find that the findings from our paper in the math domain hold on the code domain as well: (1) models with stronger solo reasoning do not consistently achieve higher recoverability, and (2) guidability is not strictly correlated with benchmark strength. Interesting, the AM-Thinking-32B, which we specially highlighted as a model with a strong benchmark scores and very poor recoverability on math reports a similar gap between on the code domain as well. This highlights an inherent weakness of this model that is not reflected in standard benchmarks but is exposed via our twin tests.
>
> Since our central findings transfer to coding tasks, we believe this addresses the generalization concern. As for creative writing task, not only is it difficult to reliably evaluate and verify without human efforts, but also most open-weight models are not trained to reason about writing.
>
> **If the reviewer finds these satisfactory, we respectfully ask that they consider raising their score to reflect these clarifications and improvements.**
>
> | Model              | Benchmark Avg. | Recoverability (Sh.) | Recoverability (Ind.) | Guidability (Sh.) | Guidability (Ind.) |
> |-------------------|----------------|-----------------------|------------------------|--------------------|---------------------|
> | R1-Qwen-1.5B       | 53.3          | 23.0 (+0)            | 17.9 (+0)             | 28.0 (+2)         | 52.8 (+2)          |
> | OpenThinker-1.5B   | 53.4          | 59.2 (+10)           | 47.6 (+10)            | 37.3 (+2)         | 53.4 (+2)          |
> | DeepScaleR-1.5B    | 58.8          | 45.9 (+4)            | 34.3 (+2)             | 27.1 (-2)         | 42.7 (-1)          |
> | DeepMath-1.5B      | 59.8          | 53.8 (+6)            | 40.6 (+4)             | 27.4 (-2)         | 42.2 (-3)          |
> | Qwen3-1.7B         | 74.7          | 70.2 (+9)            | 52.1 (+8)             | 44.4 (+0)         | 56.6 (+1)          |
> | R1-Llama-8B        | 76.0          | 50.1 (+3)            | 43.7 (+5)             | 48.2 (+1)         | 57.1 (+1)          |
> | R1-Qwen-7B         | 77.7          | 35.2 (-5)            | 27.5 (-5)             | 47.9 (-1)         | 55.7 (-2)          |
> | OpenThinker-7B     | 80.1          | 68.4 (+5)            | 61.6 (+6)             | NA                | NA                 |
> | Qwen3-8B           | 87.4          | 54.4 (+2)            | 42.2 (+0)             | NA                | NA                 |
> | Qwen3-32B          | 87.8          | 39.2 (-5)            | 31.6 (-7)             | NA                | NA                 |
> | Qwen3-30B          | 89.0          | 48.6 (-3)            | 42.8 (-1)             | NA                | NA                 |
> | R1-Qwen-32B        | 89.7          | 37.4 (-8)            | 39.0 (-5)             | NA                | NA                 |
> | AM-Thinking-32B    | 89.7          | 36.0 (-10)           | 33.7 (-9)             | NA                | NA                 |
> | QwQ-32B            | 90.5          | 42.3 (-8)            | 38.0 (-8)             | NA                | NA                 |
>
> (The +/- change in the parentheses represents in the change in rank compared to benchmark avg.)

---

> ### Author Response · Authors · 2025-11-22
> **References**
>
> - Yang, Wang, et al. "Speculative thinking: Enhancing small-model reasoning with large model guidance at inference time." arXiv preprint arXiv:2504.12329 (2025).
>
> - Wu, Tong, et al. "Effectively controlling reasoning models through thinking intervention." arXiv preprint arXiv:2503.24370 (2025).
>
> - Akhauri, Yash, et al. "Splitreason: Learning to offload reasoning." arXiv preprint arXiv:2504.16379 (2025).
>
> - Sohee Yang, Sang-Woo Lee, Nora Kassner, Daniela Gottesman, Sebastian Riedel, and Mor Geva. 2025. How Well Can Reasoning Models Identify and Recover from Unhelpful Thoughts?. In Findings of the Association for Computational Linguistics: EMNLP 2025, pages 7030–7047, Suzhou, China. Association for Computational Linguistics.
>
> - Liu, Zichen, et al. "Understanding r1-zero-like training: A critical perspective." arXiv preprint arXiv:2503.20783 (2025).
>
> - Yu, Qiying, et al. "Dapo: An open-source llm reinforcement learning system at scale." arXiv preprint arXiv:2503.14476 (2025).
>
> - Liu, Zihan, et al. "Acemath: Advancing frontier math reasoning with post-training and reward modeling." Findings of the Association for Computational Linguistics: ACL 2025. 2025.
>
> - Zelikman, Eric, et al. "Star: Bootstrapping reasoning with reasoning." Advances in Neural Information Processing Systems 35 (2022): 15476-15488.
>
> - Zelikman, Eric, et al. "Quiet-STaR: Language Models Can Teach Themselves to Think Before Speaking." CoRR (2024).
>
> - Zhou, Zhanke, et al. "Can language models perform robust reasoning in chain-of-thought prompting with noisy rationales?." Advances in Neural Information Processing Systems 37 (2024): 123846-123910.
>
> - Wan, Ziyu, et al. "Rema: Learning to meta-think for llms with multi-agent reinforcement learning." arXiv preprint arXiv:2503.09501 (2025).
>
> - Niklas Muennighoff, Zitong Yang, Weijia Shi, Xiang Lisa Li, Li Fei-Fei, Hannaneh Hajishirzi, Luke Zettlemoyer, Percy Liang, Emmanuel Candes, and Tatsunori Hashimoto. 2025. s1: Simple test-time scaling. In Proceedings of the 2025 Conference on Empirical Methods in Natural Language Processing, pages 20286–20332, Suzhou, China. Association for Computational Linguistics.
>
> - Marjanović, Sara Vera, et al. "DeepSeek-R1 Thoughtology: Let's think about LLM Reasoning." arXiv preprint arXiv:2504.07128 (2025).

---

> ### Comment · Reviewer_Y2jS · 2025-11-26
>
> Thank you for the response addressing W1 and the extensive experimentation regarding W2.
>
> Based on the following details disclosed by the authors in the rebuttal, my concerns regarding W1 have been adequately addressed. Therefore, I raise my score from 4 to 6.
>
> > The final concern raised is that our strategy results in “incoherent” sentences. In guidability, the steer is always appended at the start of the thinking, and therefore add no incoherence. In recoverability, we **keep a coherent prefix (complete sentence) of the original reasoning** and then append the distracting reasoning on a new line, **prefixed with a short transition such as"Wait. Let me think"**. Moreover, Figure 4 shows that the recoverability of the models is lowest when the distracting steers are inserted at the 0% position, i.e. at the start of the thinking which does not add any incoherence. This clearly demonstrates that poor recoverability cannot be attributed to incoherent sentences.
>
> I maintain a marginally above threshold rating due to these remaining concerns:
>
> 1. The extra details provided by the authors are not explained in the text of the paper (though the wait let me think part can be seen in Figure 1). I recommend clarifying these details in the methodology section. I also think that the overall readability of the methodology section could be improved by using plain english sentences, similar to those in the author response, and more ample use of examples. Additionally, I think the authors could better justify their methodology in the paper, by explaining  the constraints and approaches from the previous literature, mentioned in the author response.
> 2. I am not strongly convinced that the evaluation results will align with the real-world capability of these models under various collaboration methods. This is due to the fundamental constraints of this kind of analysis, mentioned by the authors.

---

> > ### Author Response · Authors · 2025-12-02
> > **Response to Reviewer Y2jS' comments**
> >
> > We thank the reviewer for acknowledging our clarifications and additional experiment results. We believe the remaining concerns are also easy to address:
> >
> > 1. We will incorporate the reviewer’s suggestions to further improve readability. In particular, we will add additional implementation details in the methodology section and better motivate the design choices in our twin-test protocol by explaining the constraints from in prior work.
> > 2. As noted above, our approach follows established practice in this line of work by stress testing models in controlled synthetic scenarios. It is important to note that the goal is not to perfectly align with the real world, but to expose models’ inherent limitations in a simualtion setup, enabling more controlled and reproducible analysis.

---

### Author Response · Authors · 2025-11-28
**Gentle Reminder. Discussion Period Ending Soon**

We thank all the reviewers for their thoughtful feedback. We hope our clarifications and additional experiments address their concerns. As the rebuttal period is nearing its end, please let us know if there are any further points you would like to discuss.

---

### Author Response · Authors · 2025-12-02
**General Comments for AC consideration**

Dear Area Chair,

We sincerely appreciate your time and efforts in handling our submission. During the rebuttal period, we actively addressed the concerns raised by reviewers with clarifications, additional experiments, and ablation analyses. For your convenience, we summarize reviewers’ concerns and our corresponding responses below.

**Only reviewer Y2jS responded to our rebuttal** before the restriction of reviewer participation (estimated 22:00 November 27, 2025) and **increased their score from 4 $\rightarrow$ 6** after reviewing our rebuttal [(link)](https://openreview.net/forum?id=hVUIguIm14&noteId=e76r0cqZtd). Although we didn’t receive response from reviewer BaPD, zPNH, BJqx during the discussion period, we believe we addressed their concerns.

All reviewers (Y2jS, BaPD, zPNH, BJqx) emphasize the novelty of our twin-test framework. BaPD and BJqx note that it is the **first systematic study** of LLMs’ robustness and adaptability in off-trajectory multi-agent reasoning. The protocol is praised for its **simplicity and reproducibility** (Y2jS, BJqx), and the experiments provide **novel insights** (Y2jS, BaPD, zPNH, BJqx) into the gap between solo and off-trajectory reasoning. Reviewers also highlight that our SFT/RL/data-filtering controls **offer actionable training guidance** (Y2jS, BaPD, zPNH) for building collaborative reasoning models.

1. **Summary of rebuttal to Reviewer Y2jS (raised their score from 4 $\rightarrow$ 6, before the news about leakage became widely public)** [(link)](https://openreview.net/forum?id=hVUIguIm14&noteId=hY9QTkjhI1):

The reviewer raised two main concerns (W1) the test design, particularly wrt introducing coherence in the trajectories, (W2) our study being limited to the math domain.

In our reponse, we clarified our test design to address W1 and added additional results for the coding domain to address W2. In response, the reviewer stated that “Thank you for the response addressing W1 and the extensive experimentation regarding W2. […] my concerns regarding W1 **have been adequately addressed**. Therefore, I raise my score from 4 to 6.” We respond to the remaining feeback from the reviewer here [(link)](https://openreview.net/forum?id=hVUIguIm14&noteId=dqnwxOOeyr)

2. **Summary of rebuttal to Reviewer BaPD** (original score - 6, did not respond before Nov27, 2025) [(link)](https://openreview.net/forum?id=hVUIguIm14&noteId=TGnB7q4E2S):

The main concern raised by the reviewer was that our paper mainly reported results on the math domain. To address this, we ran our twin-test framework on the coding domain (as explained in [link](https://openreview.net/forum?id=hVUIguIm14&noteId=pfa1hpsCgO)) and show that our findings hold and generalize well to this domain. We also explain practical reasons for why many community models cannot be evaluated against commercial models on other domains. Note that **reviewer Y2jS raised the same concern and was satisfied with our response to this, and raised their score to reflect this**.

Other concerns included whether the test design used strong distractors. In our response [(link)](https://openreview.net/forum?id=hVUIguIm14&noteId=6XZ9WS7G02), we discussed that the observed performance degradation in Table 1 already illustrates this point. The reviewer also asked for additional experiments on how results would change if the steers are sampled from a different model family (addressed in [link](https://openreview.net/forum?id=hVUIguIm14&noteId=mycwF4IRat)), examples for failure cases [(link)](https://openreview.net/forum?id=hVUIguIm14&noteId=hAG4LEO9PA), and a discussion of computational costs [(link)](https://openreview.net/forum?id=hVUIguIm14&noteId=mycwF4IRat). We believe that our responses addressed these questions adequately.

---

> ### Author Response · Authors · 2025-12-02
> **General Comments for AC consideration (Cont.)**
>
> 3. **Summary of rebuttal to Reviewer zPNH** (original score: 4, no response before Nov27, 2025) [(link)](https://openreview.net/forum?id=hVUIguIm14&noteId=TMIIvfmrSp):
>
> There are two main concerns raised by the reviewer.
>
> 1. The first concerns the generalization of our findings to non-math benchmarks, which we have adequately addressed as discussed above (for reviewers Y2jS and BaPD).
>
> 2. The reviewer requests for “mechanistinc investigations for the underlying causes for model defects (e.g., attention weights or intermediate step tracing)”. They ask to differentiate between ambiguous concepts “recognize the relevance of guidance” v/s “integrate it in reasoning process” which we/they did not define [(link)](https://openreview.net/forum?id=hVUIguIm14&noteId=TMIIvfmrSp).
>
> Based their suggestion, we conduct an attention pattern analysis and show that attention patterns can explain model failures. In particular, we show that features such as attention mass on the original (correct) question and that on (incorrect) distractor of certain attention heads are predictive of recoverability test outcomes [(link)](https://openreview.net/forum?id=hVUIguIm14&noteId=Z60ypN78wD).
>
> We also conducted a likelihood-based probe for guidability results (based on our best interpretation of the second part of their comment), which did not show a significant success vs. failure difference [(link)](https://openreview.net/forum?id=hVUIguIm14&noteId=wEMp8nHu8j).
>
> 4. **Summary of rebuttal to Reviewer BJqx** (original score: 6, no response before Nov27, 2025) [(link)](https://openreview.net/forum?id=hVUIguIm14&noteId=3wT4lHOFIi):
>
> - The reviewer raises a concern that our simulated twin-test framework might not capture the full complexity of real-world dynamics. In response, we explained that our framework is not limited to one-shot injection and can be easily extended to multi-turn settings. Moreover, our one-shot setting already exposes the critical shortcomings of existing LLMs in multi-model collaboration, which is the main goal of our study.
>
> - The second concern is still about generalizing our framework to non-math benchmarks, which we have adequetly addressed and discussed above.
>
> Other questions raised by the reviewer wrt using open-sourced datasets and the impact of RL on model performance in twin test [(link)](https://openreview.net/forum?id=hVUIguIm14&noteId=3wT4lHOFIi) have been address in our original submission, so we direct AC to the corresponding sections and experiment details in our paper [(link)](https://openreview.net/forum?id=hVUIguIm14&noteId=DBTtk1Y6sj).
>
> We believe that collectively, these additional insights address this particular concern.

---

### Meta-Review · Area_Chair_be8A · 2026-01-04

**Summary:**

Reviewers concur that the paper presents a novel and timely evaluation framework—Recoverability and Guidability—for off-trajectory reasoning, supported by systematic experimentation. They also note that several of the reported findings are counterintuitive and intellectually interesting.

The primary concerns raised by the reviewers can be summarized as follows:
- the realism and validity of the proposed evaluation protocol;
- the limited task coverage beyond mathematical reasoning;
- the lack of sufficiently deep mechanistic analysis to explain the observed failure modes.

**Reviewer Concerns:**

**Addressed concerns**

- Reviewers expressed concern about the limited domain scope due to the math-only evaluation. In response, the authors introduced additional experiments on coding benchmarks and demonstrated that the main findings remain consistent across domains.

- Reviewers also questioned whether the sampled distractors were sufficiently strong and whether the results were sensitive to sampling from the same model family. The authors addressed this by providing additional ablation studies showing similar recoverability trends with distractors drawn from different model families, along with supporting qualitative examples.

**Remained concerns**

- Although the rebuttal offered further justification and connections to prior work, concerns about the extent to which benchmark results reflect real-world multi-agent collaboration scenarios remain only partially addressed.

- While the authors included additional attention-based analyses, the explanation of how models fail to recognize and effectively integrate external guidance remains limited.

**Reviewer Scores:**

Reviewer Y2jS explicitly acknowledged that the concern regarding the math-only evaluation was adequately addressed and accordingly raised their score from 4 to 6. I expect the other reviewers to maintain their original scores. So, the expected scores are 6,6,6,4

---

### Decision · Program_Chairs · 2026-01-26

Accept (Poster)